# Knowledge-guided data mining on the standardized architecture of NRPS: Subtypes, novel motifs, and sequence entanglements

**Ruolin He**[1], **Jinyu Zhang**[2,3], **Yuanzhe Shao**[4], **Shaohua Gu**[1,4], **Chen Song**[1,4], **Long Qian**[1], **Wen-Bing Yin**[2,3]*, **Zhiyuan Li**[1,4]*

**1** Center for Quantitative Biology, Academy for Advanced Interdisciplinary Studies, Peking University, Beijing, China, **2** State Key Laboratory of Mycology, Institute of Microbiology, Chinese Academy of Sciences, Beijing, PR China, **3** Savaid Medical School, University of Chinese Academy of Sciences, Beijing, PR China, **4** Peking-Tsinghua Center for Life Sciences, Academy for Advanced Interdisciplinary Studies, Peking University, Beijing, China

* yinwb@im.ac.cn (WBY); zhiyuanli@pku.edu.cn (ZL)

⊙ OPEN ACCESS

**Data Availability Statement:** NRPS Motif Finder is an online platform for construction of NRPS motif-and-intermotif architecture (http://www.

## Abstract

Non-ribosomal peptide synthetase (NRPS) is a diverse family of biosynthetic enzymes for the assembly of bioactive peptides. Despite advances in microbial sequencing, the lack of a consistent standard for annotating NRPS domains and modules has made data-driven discoveries challenging. To address this, we introduced a standardized architecture for NRPS, by using known conserved motifs to partition typical domains. This motif-and-intermotif standardization allowed for systematic evaluations of sequence properties from a large number of NRPS pathways, resulting in the most comprehensive cross-kingdom C domain subtype classifications to date, as well as the discovery and experimental validation of novel conserved motifs with functional significance. Furthermore, our coevolution analysis revealed important barriers associated with re-engineering NRPSs and uncovered the entanglement between phylogeny and substrate specificity in NRPS sequences. Our findings provide a comprehensive and statistically insightful analysis of NRPS sequences, opening avenues for future data-driven discoveries.

## Author summary

NRPS, a gigantic enzyme that produces diverse microbial secondary metabolites, provides a rich source for important medical products such as antibiotics and antitumor agents. Despite the extensive knowledge gained about its structure and a large amount of sequencing data available, the frequent failure of re-engineering NRPS in synthetic biology highlights that much still needs to be discovered. In this work, we applied existing knowledge to data mining of NRPS sequences, using well-known conserved motifs to partition NRPS sequences into motif-intermotif architectures. This standardization allows for integrating large amounts of sequences from different sources, providing a comprehensive overview of NRPSs across different kingdoms. Our findings included new C domain subtypes, novel conserved motifs with implications in structural flexibility, and

bdainformatics.org/page?type=NRPSMotifFinder).
The source codes data can be found in S4 and S5 Files.

**Funding:** This work was supported by the National Key Research and Development Program of China (No. 2021YFF1200500 to ZL), National Natural Science Foundation of China (No. 32071255 to ZL, No. 42107140 to SG), Clinical Medicine Plus X - Young Scholars Project, Peking University, the Fundamental Research Funds for the Central Universities (No. PKU2022LCXQ009 to ZL), The Biological Resources Program, Chinese Academy of Sciences (KFJ-BRP-009-005 to WBY), Key Research Program of Frontier Sciences, Chinese Academy of Sciences (ZDBS-LY-SM016 to WBY) and National Postdoctoral Program for Innovative Talents (No. BX2021012 to SG). The funders had no role in study design, data collection and analysis, decision to publish, or preparation of the manuscript.

**Competing interests:** The authors have declared that no competing interests exist.

insights into why NRPSs are so difficult to re-engineer. To facilitate researchers in related fields, we constructed an online platform, "NRPS Motif Finder", for parsing the motif-and-intermotif architecture and C domain subtype classification. We believe that this knowledge-guided approach not only advances our understanding of NRPSs but also provides a useful methodology for data mining in large-scale biological sequences.

## Introduction

Non-ribosomal peptide (NRP) synthesized by non-ribosomal peptide synthetase (NRPS) are a diverse family of natural products widespread in fungi and bacteria [1–5]. According to the BiG-FAM database (as of 2022/8), NRPS is the most prevalent class (29.30%) out of the 1,060,938 biosynthetic gene clusters (BGCs) detected from 170,585 bacterial genomes, and 46.81% out of the 123,939 BGCs from 5,588 fungal genomes [6]. Microbes utilize NRP for various niche-construction activities, such as communication [7,8], defense [9–13], and resource-scavenging [14]. Diverse NRPs serve as reservoirs for pharmaceutical innovations, with anti-cancer, anti-bacterial, anti-fungal, anti-viral, cytotoxic, and immunosuppressive activities [15–18]. To date, nearly thirty NRP structural medicines have been approved for commercialization [15], including actinomycin [19], penicillin [20,21], and vancomycin [22].

Many re-engineering efforts have been inspired by the pharmacological potentials of NRPS and its modular architecture re-engineering [23,24]. NRPS synthesizes peptides in an assembly-line fashion using its repeating module units. Each module is generally comprised of three key domains, including the adenylation (A) domain, which recognizes the substrate and activates it as its aminoacyl adenylate; the condensation (C) domain, which catalyzes amide bond formation between two substrates; and the thiolation (T) domain, which shuttles the substrates and peptide intermediates between catalytic domains. Other optional domains are also important, such as the thioesterase (TE) domain or terminal condensation-like ($C_T$) domain [25], which cyclize or release peptides from the NRPS and generally occurs at the end of an NRPS; epimerization (E) domain, which catalyzes the conversion of L-amino acids substrate to D-amino acids and usually follows the C domain in case it occurs.

In order to create novel products, many synthetic efforts have been made to re-arrange these building blocks of NRPSs. For example, due to the central roles of A domains in substrate recognition and activation, early works focused on the A domain, including A domain substitution, reprogramming A domain substrate coding residues, and A subdomain substitution [23]. Later, considering the possible roles of C domains in substrate selectivity, researchers attempted to re-engineer catalytic domains together with the A domain, such as substituting a complete module, or the C+A domain and T+C+A domain [23]. Recently, Bozhüyük et al. advanced the exchange unit concept and recombined NRPS at a fusion point between the C and A domains which they named as "C-A linker" [26]. They also found another cutting point inside the C domain [27]. Calcott et al. also attempted to recombine at another cutting point between the C and A domains [28], almost at the start of the C-A linker defined by Bozhüyük et al. However, in most cases, re-engineering often substantially reduces the product yield. Furthermore, conversions of these units were typically performed between two substrates with similar chemical properties, based on well-studied NRPS systems with a small number of sequences [29,30]. A systematic understanding of the rational design of NRPS has yet to be developed.

The massive expansion of microbial sequencing data has created opportunities for a comprehensive understanding of NRPSs [31]. As of October 2021, the Integrated Microbial

Genomes Atlas of Biosynthetic gene Clusters (IMG-ABC) v5 database contains a total of 411,027 BGCs based on sequence annotation [32], and Minimum Information about a Biosynthetic Gene Cluster (MiBiG) v2 database contains 1926 BGCs confirmed to be biologically active [33]. Together with the accelerated expansion of databases, bioinformatics platforms that specifically predict and annotate the secondary metabolites are also in rapid development [34], such as antiSMASH [35], PRISM [36], and SeMPI [37]. Pfam, a widely utilized database of protein families and domains since 1997 [38,39], has also been commonly used for NRPS domain detection by profile hidden Markov models (HMM). They have been used to aid in the discovery of new NRPS antibiotics [16,40]. However, different bioinformatics tools and platforms employ different annotation standards and algorithms, and their definitions of domains and inter-domains in NRPS need to be clarified.

Aside from annotating the module-domain architecture, predicting the substrate specificity of A domains in NRPS has been a focus of research for many years, with various algorithms being developed to address the challenge [37,41–44]. Some use supervised learning and rely on a training set of full-length A domain sequences and their known substrates [42]. Other well-known approaches incorporate structural information of the A domain [45]. With advances in protein structures, the substrate-specifying residues of modular mega-enzymes, such as NRPS [29,46] and polyketide synthase (PKS) [47], were identified. Based on the first crystal structure of the A domain (PheA, PDB 1AMU) [29], Stachelhaus et al. identified ten substrate-specifying residues in the A domain's binding pocket. These ten residues, known as the "specificity-conferring code", strongly associate with the substrate specificity, and have been widely exploited by researchers to improve prediction efficiency [45]. However, the diversity of the specificity-conferring code for the same substrate remains challenging, making the performance of current machine learning algorithms heavily dependent on the training set. Actually, diverse forms of the specificity-conferring code could exist for the same substrate [45]. Such diversity of the specificity-conferring code causes barriers in extending the prediction algorithm: without a one-to-one mapping between the specificity-conferring code and substrate specificity, the performance of current machine learning algorithms heavily relies on the training set. Indeed, the performances of these algorithms in fungal NRPSs are unsatisfactory, as fungi have a significantly smaller number of A domains with known substrates. Furthermore, re-engineering based on modifying the specificity-conferring code does not operate as expected [46], highlighting the complexities in the A domain's function.

Coevolution analysis has been used as a general strategy to uncover protein interactions and functional couplings from a large number of sequences [48]. Existing methods in coevolution analysis include normalized mutual information (MI) of amino acid frequencies [49], direct coupling analysis (DCA) [50], protein sparse inverse covariance (PSICOV) [51], and statistical coupling analysis (SCA) [52,53]. Among them, SCA has been utilized to identify appropriate boundaries for protein re-engineering [54]. The basic idea of SCA is to identify groups of residues sharing the same modes of covariations, referred to as "sectors" [55]. Residues projecting to a single sector can be viewed as "evolving together", and are likely to cooperate in function [56]. Therefore, sectors could serve as the "evolutionary units" for re-engineering, as the borders of sectors provide rational cutting-points for preserving functional connections between residues [55].

Applying SCA to NRPS sequences to gain insights on re-engineering is a straightforward idea. However, one obstacle to this simple notion is the need for manually curated sequences: the majority of sequences in databases were automatically annotated by machine learning algorithms, and have yet to be reviewed by human experts. Moreover, different algorithms with varying standards for annotation were adopted by different researchers, which muddled the exact definitions of domains and interdomains in NRPSs. For example, A and C domains

annotated by the default option of antiSMASH are substantially shorter than Pfam-annotated domains [38], and are usually different from sequences from the Protein Data Bank (PDB) [57–59]. Even the domain annotations from antiSMASH, one of the most popular annotation platforms for NPRSs, slight changes happened between versions.

One solution for this "confusion by the annotation" is to focus on the invariant sequence pieces in NRPS architecture, known as "core motifs" [60]. Despite the enormous diversity in NRPs, highly conserved sequence elements have been discovered across domains, and are usually essential to the domain function. These well-established core motifs include: 7 motifs in the C domain, 10 motifs in the A domain (of note, these are not the specificity-conferring code), 7 motifs in the E domain, 1 motif in the T domain, and 1 motif in TE domain [60]. However, these motifs have been proposed for more than two decades. Since then, these motifs have yet to be systematically updated or confirmed. These conserved motifs could be used to standardize NRPS domains, allowing for statistical analysis of NRPS sequences from large databases without manual curation.

In our work, we constructed a "coordinate system" for NRPSs based on well-known conserved core motifs. Such motif-and-intermotif architecture allowed us to integrate sequence information of NRPSs from multiple databases and extract global statistics on the length, variation, and functional properties of each NRPS sequence. Following that, we were able to systematically evaluate a large number of sequences for their statistical properties and function-sequence couplings. For example, we identified new C domain subtypes and obtained the most comprehensive cross-kingdom classification of C domains to date. In addition, we uncovered several previously unrecognized conserved motifs, which could have functional implications and serve as candidates for novel core motifs. One of these novel motifs is located close to the A domain's specificity pocket with implications in structural flexibility, and mutation experiments supported its importance. Then, using coevolution analysis, we dissected two significant challenges in re-engineering NRPSs: first, there is no cutting point to separate multiple overlapping coevolving sectors, limiting domain/subdomain swapping; second, the specificity of the A domain intertwines with not only the specificity-conferring code but also the length of five loops in the A3-G intermotif, complicating the rational modification of A domain. Our findings provide a comprehensive overview and statistical insights into the sequence features of NRPSs, paving the way for future data-driven discoveries. In the end, an online platform entitled "NRPS Motif Finder" is offered for parsing the motif-and-intermotif architecture and C domain subtype classification (http://www.bdainformatics.org/page?type=NRPSMotifFinder).

## Result

### 1. Overview of the motif-and-intermotif architecture of NRPS

**1.1. Sequence properties of motifs and intermotifs in NRPS domains.** To establish the foundation of our study, we first partitioned the NRPS architecture based on well-known core motifs. The sequence representations of 26 well-studied motifs were curated from the literature [60], then located to 7,329 NRPS domains from MiBiG database by alignment (see Method for details). As expected, most known core motifs are highly conserved, exhibiting clear sequence logos (first panel, Figs 1A–1D and S1–S5). Motifs from the A, E, and T domains have sequence identities higher than 60%. However, C domain motifs exhibit a lower sequence identity than A, T, and E domains. Among the C motifs, motif C3 is the most conserved, with 55% sequence identity across all C subtypes (Fig 1A, first panel). High conservation of C3 is reasonable due to its critical role in catalysis [61,62]. Except for the Heterocyclization subtype, where the motif C3 is DxxxxD, the motif C3 in other subtypes is almost always HHxxxD.

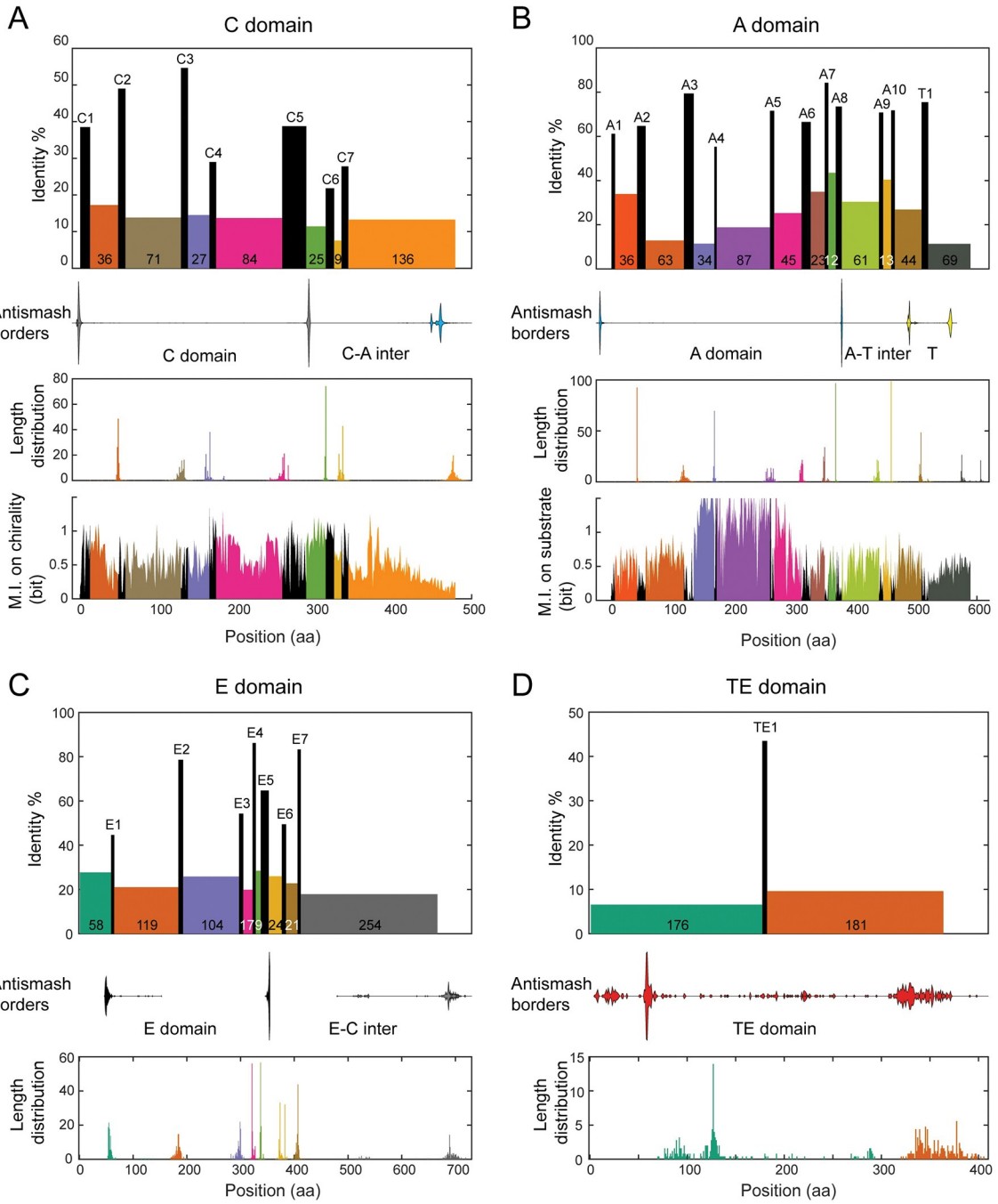

**Fig 1. Overview of the motif-and-intermotif architecture of canonical NRPS domains. A.** Organization of C domain. Rectangles in the first panel show the seven C motifs (black) and eight inter-motif regions (colored), including the last one as the inter-motif region between the C7 and A1 motifs. The heights of the rectangles indicate the average sequence identities. The widths of the rectangles are their average lengths in amino acid sequences, with the average lengths of inter-motif regions printed on their rectangles. Violin plots in the second panel mark the border positions of domains annotated by antiSMASH 5.1 by the default option (the start and end borders of the C domain: gray; the start of the A domain: blue). The third panel shows the distributions of the ending positions of the eight inter-motif regions, counting from their preceding conserved motifs. The fourth panel shows the mutual information (M.I.) between residues in the C domain and chirality subtypes. **B.** Same as that in **A**, but for the A and T domains. It starts with the A1 motif and ends before the C1 motif. In the violin plot, the start and end borders of the A domain are colored blue; the start and end borders of the T domain are colored yellow. The fourth panel shows the mutual information between residues and A domain substrate specificity (see Method for details). **C.** Same as that in **A**, but for the E domain. It starts after the T1 motif, and ends before the C1 motif. In the violin plot, the start and end borders of the E domain are colored black; the start of the C domain is colored gray. **D.** Same as that in **A**, but for the TE domain. It starts after the T1 motifs, and ends with regions after the annotated TE domain. In the violin plot, the start and end borders of the TE domain are colored red.

Motifs C6 and C7 are less conserved, 22% and 28% identities, respectively. This lower conservation in C motifs is likely due to various functional subtypes within the C domain [61], which we will discuss in further detail in the following sections 1.3–1.5.

The sequence region between the conserved motifs, referred to as the "intermotif region", has a comparatively lower level of sequence identity, with values ranging from 10%-25%. Most intermotif regions display narrow length distributions, with a standard deviation smaller than 10 aa (third panel, Fig 1A–1C), demonstrating the architecture invariance of NRPSs. However, the TE motif at the end of the cluster exhibits a much wider distribution (320-380aa, Fig 1D). An additional noteworthy exception is observed in the intermotif region between the motif T1 and its next motif C1, which exhibits a bi-modal distribution (Fig 1B). Upon closer examination of the domain annotations, it was found that the length of the T1-C1 region depends on the MiBiG annotation of the following C domain (S3B Fig): Specifically, regions followed by the LCL subtype tend to be 65–70 amino acids in length, whereas regions followed by the Dual subtype are typically 95–105 amino acids in length. These observations suggest that the architecture of intermotif regions may have functional implications.

It's worth mentioning that by default, antiSMASH annotations for the C domain do not include the motif C6 and C7; these for the A domain do not include the motif A9 and A10; and these for the E domain do not include the motif E6 and E7 (second panel, Fig 1A–1C). We noted the motif E6 and E7 are located in the TIGR01720 domain annotated by antiSMASH. This explains the difference in NRPS domain length between MiBiG and Pfam (S6 Fig).

**1.2. Exploring the coupling between sequence and function in the C and A domains by mutual information.**   Another focus of this study is how sequences couple with functions. Conserved motifs serve as "anchors" in multiple sequence alignment, facilitating a detailed evaluation of information content. Given that the sequences of the C domain have been previously demonstrated to be indicative of chirality subtypes [61], we first quantified the Shannon mutual information (MI) between the aligned sequences and functional subtypes in the C domain, residue by residue (see Method for details). The region between the C1 and C7 motifs, represented in the last panel of Fig 1A, contains a substantial amount of information about chirality subtypes. After the C7 motif, the information about chirality gradually decreases until it reaches the baseline level as the sequence progresses toward the A1 motif (Fig 1A, last panel).

Generally, conserved motifs contain low levels of mutual information (Fig 1). Nevertheless, it's noted that the C5-C6 region, including the C5 and C6 motif, has a high MI value with C domain subtypes (Fig 1A, last panel). We checked this region in the structure of LgrA, the only structure that contains multi-module NRPS in multiple conformations so far characterized by Reimer et al.[63]. We noticed that this region is located at the interface between the donor T domain and the recipient C domain of the next module ($T_n$:$C_{n+1}$), which has been suggested to play roles in chirality [64]. Multiple coevolution positions between the $T_n$ domain and $C_{n+1}$ domain were suggested by DCA in Reimer et al.'s work, with multiple key residues located in the C5 and C6 motif. Therefore, $T_n$ and $C_{n+1}$ are functionally linked by the C5-C6 motif in the $T_n$:$C_{n+1}$ interface, which explains the high MI value between C5-C6 intermotif region and C domain subtypes.

In light of the critical role played by the A domain in substrate selectivity, we also employed the same mutual information measurement to evaluate the relationship between each position in the A domain and substrate specificity (see Method for details). Results in the last panel of Fig 1B show that positions with high information content start from the A3 motif and end before the A6 motif. This region overlaps the substrate's binding pocket [57]. It is worth noting that the A4-A5 region has been well-known for containing the specificity-conferring code [29].

**1.3. Cross-kingdom validation and comparison of motif properties in a larger dataset.** The MiBiG database provides a solid foundation for discovering the fundamental properties of NRPSs, as all BGCs in the database have been confirmed to be active. Then, to further validate the conservation of the NRPS motifs, we expanded our analysis to a larger dataset. With that aim in mind, we downloaded all complete bacterial genomes (30,984) and all assembly-level (including "contig", "scaffold", "chromosome", and "complete" levels) fungal genomes (3,672) from NCBI. After removing genomes with 99.6% similarity, we had 16,820 bacterial and 2,505 fungal genomes (as of 2022/10/23). BGCs in these genomes were annotated using antiSMASH v6. Unlike the MiBiG database, where the majority of pathways have been experimentally verified to be active, an unknown number of the domains in this dataset might be inactive. To account for this, we employed a preliminary dead domain filter by eliminating domains with non-standard motif lengths (see Method for details). In this large dataset, we analyzed 83,489 C domains, 95,582 A domains, 86,688 T domains, 14,502 E domains, and 23,590 TE domains from bacteria; and 34,269 C domains, 40,458 A domains, 26,651 T domains, 3,982 E domains, and 4,008 TE domains from fungi.

The prevalence of NRPS pathways extends across multiple kingdoms, with fungi being the only known eukaryotic producers. Examining the similarities and differences in the NRPS sequences of bacteria and fungi can help elucidate the conservation and divergence of NRPS functions. In contrast to the MiBiG database, where most pathways are of bacterial origin, the larger dataset contains a comparable number of C, A, and T domains from both fungi and bacteria, providing a valuable platform for validating and comparing motif properties.

Overall, the motif logo and intermotif length of the A, T, E, and TE domains were found to be similar between bacteria and fungi, albeit with some subtle differences (S7–S14 Figs). For example, the sequence logo of bacteria E6 was identified as RX(V/I/L)PXXGXG(Y/F)G, while fungi E6 was RX(V/I/L)PXXGXXYF (S11 Fig); the fungal intermotif A10-T1 was found to be 13 amino acids longer than that of bacteria by the median (S13 Fig); the fungal intermotif E2-E3 is 15 aa shorter than bacteria by the median (S14 Fig). The disparity in the E6 motif highlights a potential bias in our current understanding of NRPS, which is predominantly informed by sequences collected from bacteria: Previous research reported that the E6 consensus sequence was PXXGXGYG, only consistent with the E6 sequence logo we observed in bacteria [60]. A more comprehensive understanding of NRPS motif and intermotif across kingdoms is required to expand our knowledge.

**1.4. Identification of new C subtypes in fungi guided by conserved motifs.** The C domain of NRPS exhibits notable differences between bacteria and fungi. Previous research has established that the C domain can exist in multiple subtypes, each displaying distinct conserved motifs [61,65,66]. In order to conduct a comprehensive comparison of NRPS between fungi and bacteria, it is imperative to first clarify these subtypes. Currently, antiSMASH v6 can distinguish LCL, DCL, Cglyc, Dual, Starter, Cyc, and X subtypes [35], and NaPDoS v2 can distinguish LCL, DCL, Dual, Starter, Cyc, modAA, and Hybrid subtypes [67]. However, these tools are primarily based on bacterial subtypes and do not consider fungal subtypes such as CT and Iterative [25,68]. This lack of annotation for fungal C domain subtypes hinders genome mining of fungal NRPS.

In this study, we curated sequences representing 18 subtypes of the C and E domains, 14 of which were sourced from the conserved domain database (CDD) [69], and the remaining 4 were obtained from the literature [33,66,70,71] (S1 Table). Phylogenetic analysis of these subtypes revealed their relatedness (Fig 2A). These subtypes form two major clades: L-clade and D-clade, consistent with previous reports [65]. Furthermore, our results revealed the evolutionary landscape of all known C-domain subtypes for the first time, offering deeper insights into several sub-clades. It can be observed that, the CT subtype may evolve from the FUM14

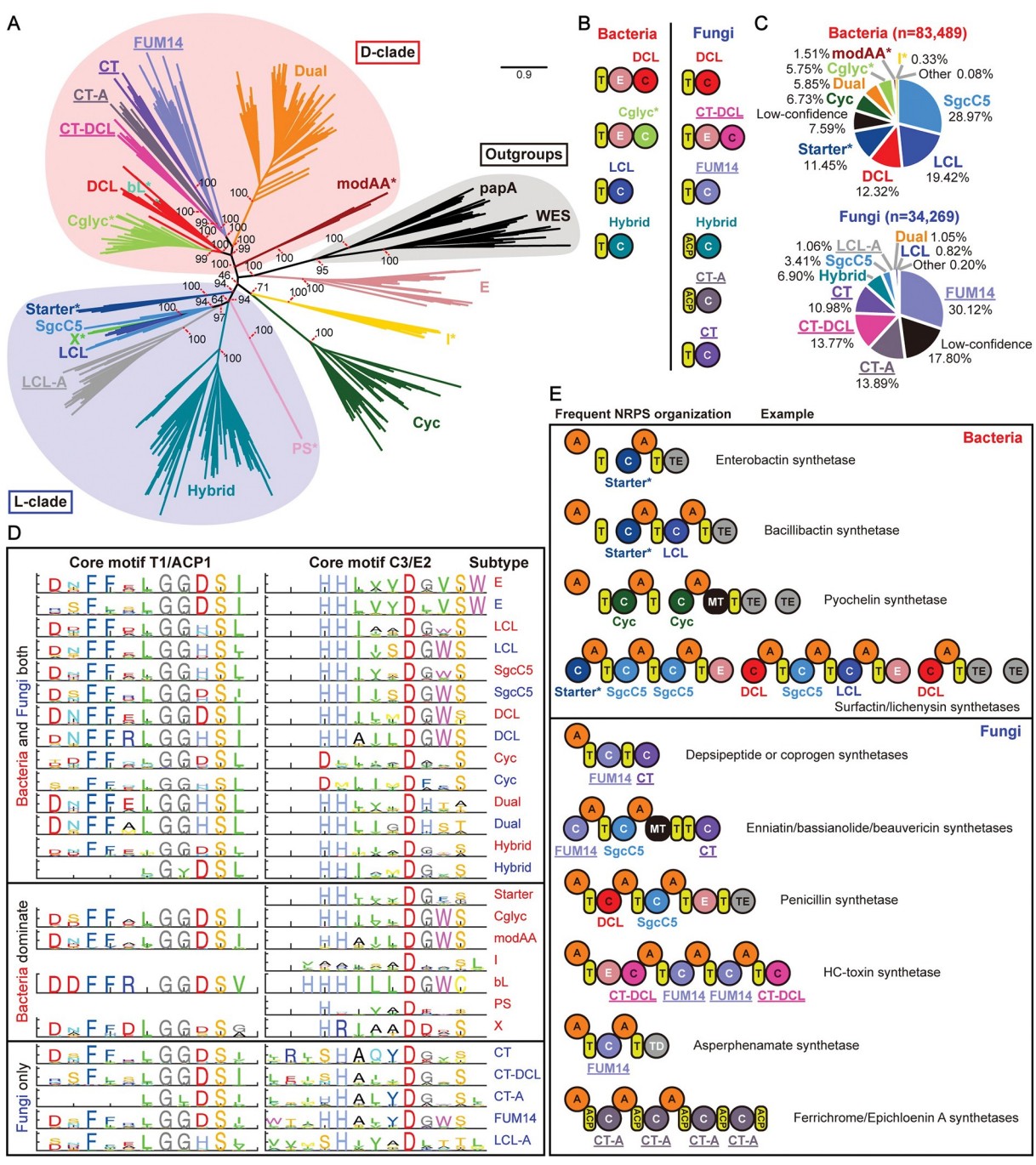

**Fig 2. C domain subtype analysis and representative NRPS organizations in bacteria and fungi. A.** Maximum-likelihood phylogenetic tree of the condensation domain superfamily. Subtype classification and sequences are described in the main text and the Method. Different subtypes are indicated by colors, with subtypes exclusive to fungi marked by underlines, and subtypes found predominantly in bacteria marked by asterisks. This tree is rooted, taking papA and WES as outgroups [65] (black shading). L-clade and D-clade are indicated by blue and red shading, respectively. **B.** Domains adjacent to different C domain subtypes in bacteria and fungi. **C.** The statistics of subtype distribution in 83,489 bacterial C domains and 34,269 fungal C domains. C domains with HMM scores above the empirical threshold of 200 were annotated by their predictions, otherwise marked as "Low-confidence". **D.** The sequence logo for the C3 or E2 motif from different C domain subtypes and the T1 or ACP1 motif adjacent to each subtype. Sequences from bacteria were marked by red, while sequences from fungi were marked by blue. **E.** Frequent NRPS organizations with known representative examples in bacteria and fungi.

subtype; the LCL subtype may evolve from SgcC5, and Cglyc may evolve from DCL. The FUM14 subtype was named from *Fusarium verticillioides* FUM14 (also known as NRPS8), a bi-domain protein with an ester-bond forming NRPS C-domain [72]. SgcC5 is a free-standing NRPS condensation enzyme (rather than a modular NRPS), which catalyzes the formation of both ester and amide bonds [73]. Cglyc subtype is the glycopeptide condensation domain [61].

New insights on subclasses in C domains can be obtained by precisely characterizing this C domain phylogeny, particularly in fungal-related domains. For each C domain subtype, we constructed MSA from curated sequences by Muscle v5 [5] and built reference HMM models by hmmer v3 [74]. We then utilized these HMM models to classify C domain subtypes, and extracted motifs for each subtype by the previously mentioned method.

During this process, we noticed that the sequence motifs of the curated CT subtype are remarkably variable, suggesting that this "CT" may not be a uniform subtype. Guided by the motif sequence and phylogenetic analysis, our analysis suggests that the original "CT" domain can be further classified into three subtypes by sequence similarity, which is also highly related to their locations in the NRPS pathway (Fig 2B): One of the subtypes is always located at the end of the NRPS, and we termed it "CT" because it fits the original definition of the CT subtype. Another subtype was termed "CT-DCL" because it is always behind an E domain and may function similarly to the DCL in bacteria. Of note, we also observed that the annotated fungal DCL subtype is always after the T domain. The third CT subtype is atypical because it is always behind an ACP (acyl carrier protein) domain rather than a T domain. Both the ACP and T domains are phosphopantetheinyl carrier domains, but the ACP domain has a conserved XGXDSL motif rather than the T domain's GG(D/H)S(I/L) motif [60,75]. Therefore, we termed this subtype "CT-A" (meaning atypical CT). These three CT subtypes form three different clades in the phylogenetic tree.

After clarifying subtypes, the overall statistics regarding the distribution of C domain subtypes in bacteria and fungi can be analyzed. Our subtype prediction covered 92.41% C domains in bacteria and 82.20% C domains in fungi, by an empirical score threshold of 200 in HMM models (Fig 2C). Our predictions for the Starter and Dual subtypes are consistent with antiSMASH. However, antiSMASH frequently assigns incorrect annotations of "LCL" and "DCL" to subtypes that are uncommon in bacteria, which were corrected in our annotation (S2 Table).

By the updated subtype classification, we redrew sequence logo of C domain in MiBiG. The conservations of motifs increased significantly compared to the prior classification relying on MiBiG (S1 and S15 Figs). Regarding subtype distribution, bacterial C domains mainly consist of SgcC5, LCL, DCL, Starter, Cyc, Dual, and Cglyc, while fungal C domains are mainly composed of FUM14, CT-A, CT-DCL, CT, and Hybrid. Among them, I (interface) only existed in bacterial C domains. Starter, Cglyc and modAA are almost exclusively found in bacteria, with a handful of occurrences in fungi (S3 and S4 Tables). FUM14, CT-A, CT-DCL, CT, and an atypical LCL (designated as LCL-A, meaning atypical LCL which usually has a non-canonical conserved SHXXXDXX(T/S), rather than HHXXXDGXS) only exist in fungi.

These distinctions between fungal and bacterial C domains and differences between subtypes are readily apparent in the comparison of typical motif logos (S16 and S17 Figs), such as the C3 and T1 (Fig 2D). The C3 motif is essential for catalysis, and the T1 motif has been suggested to exhibit covariation with its preceding C domain [65]. Our findings highlight the variation in the location and motif of C domains with the T domain between subtypes in fungi and bacteria (Fig 2D). Firstly, some C domain subtypes do not directly precede a T domain: the fungus Hybrid and CT-A subtypes are adjacent to an ACP domain rather than a T domain; the bacteria DCL and fungus CT-DCL subtypes are located after an E domain, consistent with previous reports [65]; For the Cyc subtype, only 29.34% of them are adjacent to a T domain,

while the rest are located at the start of the pathway. The Starter, I, and PS subtypes are also exclusively found at the start of the pathway. Secondly, we observed that the coupling between the C domain and its adjacent T domain cannot be solely explained by the L- and D-clades. Previous research mainly based on bacterial data suggested that the LCL subtype (located in the L-clade of the phylogenetic tree in our analysis) is adjacent to a T domain with the LGGHSL motif, and the DCL subtype (located in the D-clade of the phylogenetic tree in our analysis) is adjacent to a T domain with the LGGDSI motif [64,65]. Our analysis not only confirmed these relationships but also showed that not all subtypes in the L- and D-clades follow this pattern: The X subtype in the L-clade is adjacent to a T domain with the LGGDSG motif; in the D-clade, and the Dual and fungal DCL subtypes in the D-clade are adjacent to T domains with the LGGHSI and LGGHSL motifs, respectively. These observations demonstrate that the T1 motifs are primarily linked to the specific subtypes of their adjacent C domains, rather than their clades.

In addition to the conserved motifs, the highly variable intermotif regions also differentiate fungi and bacteria based on some of their lengths. In the standardized motif-and-intermotif architecture, we compared the intermotif length distribution in different C domain subtypes between bacteria and fungi (S18 Fig). The most notable difference is in the T1/ACP1-C1 region. C domain subtypes with more complex functions, such as Dual (92aa), modAA (74aa), bL (82aa), CT (86aa), CT-A(82aa), and FUM14 (85aa), have a substantially longer T1-C1 intermotif compared to C domains with simpler functions, such as LCL/ScgC5 (60aa in bacteria, 59aa in fungi). This finding implies that longer interdomain length may provide the necessary space for coordination between different domains in the megasynthetase. This distinction in C domain subtype compositions and intermotif lengths may offer potential applications in the future, allowing us to distinguish between bacterial and fungal NRPS in fragmented metagenomic sequencing data.

**1.5. The prevalent NRPS organizations in bacteria and fungi.**   This detailed annotation of C domain subtypes allows us to evaluate how domains are organized within NRPS pathways in bacteria and fungi. In this work, "organization" was defined as the composition and arrangement of domains within an NRPS pathway. For the accuracy of statistics, we focused on 82.47% (12,364/14,992) bacterial and 61.00% (6,438/10,555) fungal NRPS pathways that only contain the high-confidence C domains. Our findings show that in bacteria, 24 out of the 30 most frequent organizations are involved in the production of siderophores, accounting for 28.21% of the 12,364 pathways. Among these 24 siderophore pathway organizations, enterobactin [76] (BGC0002476), the siderophore with the highest binding affinity to iron, has the highest occurrence (21.35% of 12,364). The second most frequent siderophore NRPS is bacillibactin synthetase [77] (BGC0000309), making up 2.55% of the bacterial pathways, with an organization similar to enterobactin synthetase but with an additional module containing LCL subtype C domain. The third most frequent siderophore NRPS is pyochelin synthetase (1.16%, BGC0000412), containing two Cyc subtype C domains [78]. In addition to siderophores, biosurfactant surfactin/lichenysin synthetases are also frequent [79,80] (BGC0000433 and BGC0000381), representing 2 out of the 30 most frequent organizations and accounting for 1.24% of all pathways. This biosurfactant is comprised of three NRPS genes, with 1 Starter, 2 DCL, and 4 LCL/SgcC5 C domains. Interestingly, both pyochelin synthetase and surfactin/ lichenysin synthetases contained 2 TE domains, with the function of the second TE domain suggested for proofreading for the NRPS product [81].

In fungi, the most frequent NRPS organization is also related to siderophore NRPS. Of the 20 most frequent NRPS organizations, 8 are depsipeptide or coprogen synthetases [68] (BGC0001249), accounting for 11.90% of the 6,438 pathways. These synthetases typically consist of one NRPS gene with one FUM14 and one CT subtyped C domains, with some having

an additional NRPS gene with an A domain (Fig 2E). These depsipeptide or coprogen synthetases are known for their iterative features, and their pathway organization suggests that the FUM14 subtype could also participate in the iterative process. The second most frequent siderophore NRPS is Ferrichrome/Epichloenin A synthetase [82,83] (0.79% of the pathways, BGC0000901/BGC0001250, Fig 2E). It's noted that all subtypes of the C domains in all Ferrichromes (type I-VI) are CT-A, except for the first C domain in several cases [82]. Therefore, detecting Ferrichrome-type siderophore in the genome is possible simply by C domain subtype. The other two frequent NPRS organizations ranked within the top 20 are HC-toxin and Asperphenamate synthetase (1.29% and 0.87% of the pathways, BGC0001166 and BGC0001517, respectively, Fig 2E). HC-toxin is a virulence factor produced by plant pathogenic fungi [84], and Asperphenamate exhibits antitumor activity [85]. It's noted that Asperphenamate synthetase ends with a TD domain, one of the systematic differences between bacteria and fungi [86]. The representation of the third pathway is unknown, and the most similar in MiBiG is cyclo-(D-Phe-L-Phe-D-Val-L-Val) annotated by antiSMASH v6 (BGC0000357). Other well-known fungal NRPS include enniatin/bassianolide/beauvericin synthetases [87–89] (29th most frequent organization, 0.36% of the pathways, BGC0000342, BGC0000312 and BGC0000313), which are also known for their iterative feature and have been the focus of re-engineering for new products [90,91]. These synthetases consist of one NRPS gene with one FUM14, one SgcC5, and one CT C domain, again suggesting FUM14 subtype C domain may have an iterative function. Another known pathway (34th, 0.33%) is the penicillin/isopenicillin N/benzylpenicillin/phenoxymethylpenicillin synthetase pathway [92] (BGC0000404, BGC0000405), which has a unique organization with its DCL located behind a T domain rather than an E domain, and its E domain positioned before a TE domain instead of a C domain. Almost all DCL in fungi have a similar organization to penicillin synthetase.

Interestingly, in fungi, the function of DCL in bacteria might be replaced by CT-DCL. For example, the HC-toxin synthetase features an E+CT-DCL arrangement (Fig 2E). In the entire database, such E+CT-DCL arrangements are common, with 66.14% (3121/4719) CT-DCL located behind an E domain, and 78.38% of E domain proceeding a CT-DCL (3121/3982). And in some cases, such as HC-toxin synthetase (Fig 2E), fungal NRPS ends with a CT-DCL domain. It suggests that the CT-DCL domain may act as a CT domain to terminate the synthesis of NRP. The results highlight the limitations of current annotations in large dataset, despite that the large dataset provided insights for confirming previously discovered information. Therefore, after analyzing this large dataset, we shifted our focus back to the well-studied and annotated MiBiG database, where the well-established annotation offers a more suitable platform for in-depth discovery.

## 2. Discovery and functional identification of new conserved motifs

The presence of conserved motifs acts as anchor points in multiple sequence alignments, enabling an in-depth examination of the level of conservation across the entire sequence. From the MiBiG database, we calculated the amino acid frequency and gap frequency in a sequence alignment of 1,161 C+A+T NRPS modules (1053 from bacteria, 81 from fungi, and 27 from "others"). The results are presented in Fig 3A, with a lower panel showing the conservation of the alignment. As expected, most highly conserved positions are found within or close to known core motifs, and thus can be seen as the extension of known motifs (S19 Fig). However, three highly conserved regions are relatively distant from well-known motifs (Fig 3A, upper panel), therefore, cannot be considered extensions of these well-known motifs, suggesting potential new motifs.

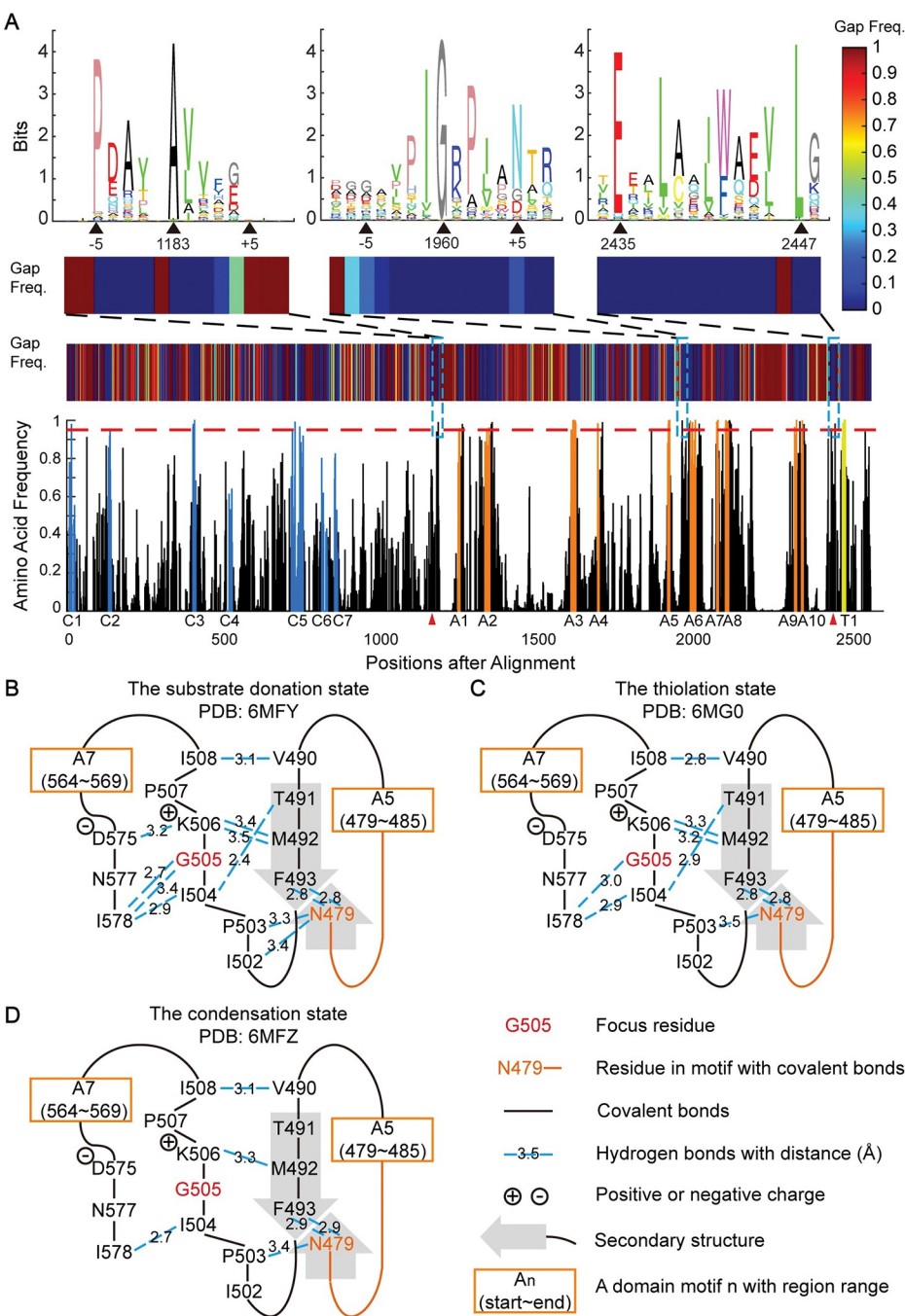

**Fig 3. Analysis of amino acid frequency reveals potential new motifs with implications for structural flexibility. A.** Amino acid frequency and gap frequency along the multiple sequence alignment of the NRPS C+A+T modules. In the bottom panel, bar heights indicate the frequency of the most frequent amino acid. Bars in the known core motifs from the C, A, and T domains were colored blue, orange, and yellow, respectively. The horizontal red dashed line represents the 0.95 frequency level. Domain boundaries annotated by Pfam are divided by red triangles. The colored patch above the amino acid frequency indicates gap frequency. Three potential new motifs (position 1183, 1960, and 2435/2447 in MSA) are marked by the blue dashed box. The upper panel shows the sequence logo and the gap frequency near the three potential new motifs. **B.** Chemical interactions and secondary structures surrounding the second potential new motif in (A) at the substrate donation state (PDB: 6MFY). Hydrogen bonds near the most conserved Gly were shown in blue dashed lines. Covalent bonds were shown as black lines. Secondary structures, such as beta-sheets, were demonstrated as bold gray arrows. Known A domain motifs adjacent to related residues were shown in the orange box. **C.** Same as that in **B**, but in the thiolation state (PDB: 6MG0). **D.** Same as that in **B**, but in the condensation state (PDB: 6MFZ).

**2.1. The new "G-motif" in the substrate binding pocket relates to conformation flexibility.** A conserved glycine located in the middle of the A5-A6 intermotif region has caught our attention. We also observed this conserved glycine in the A domain MSA in the large dataset (S7 and S10 Figs). It is 21(±2) aa from the last residue of the A5 motif, and 23(±0.2) aa from the first residue of the A6 motif. This glycine is highly conserved in over 99% of the aligned A domains, with surrounding residues being moderately conserved (as shown in the second logo in the upper-most panel of Fig 3A). Notably, this glycine divides the A5-A6 intermotif region into two halves regarding substrate information, where the sequence preceding this glycine contains substantially more mutual information about substrate (S20 Fig). For convenience, we will collectively refer to this conserved glycine and its surrounding residues as the "G-motif".

We then located this G-motif to the G505 position in a recently solved NRPS structure, the linear gramicidin synthetase subunit A (LgrA) [63]. Interestingly, the G-motif resides on a loop structure, which is typically considered non-conserved. The LgrA structure has been solved with three function-related states: 6MFY, the substrate donation state; 6MG0, the thiolation state; and 6MFZ, the condensation state [63]. By calculating hydrogen bonds in these structures, we noticed that the G-motif (comprising residues I502, P503, I504, G505, K506, P507, and I508 in LgrA) interacts with N479 in the A5 motif (where the A5 spans N479-E485, with sequence NGYGPTE) and its backside residues (V490, T491, and M492). It also interacts with the backside residues of the A7 motif (D575 and I578; where the A7 spans Y564-R569 with sequence YRTGDR) (Fig 3B–3D). Of note, the number of hydrogen bonds for these interactions changes remarkably in different function-related states (Figs 3B–3D and S21): In the substrate donation state 6MFY, the G-motif has 12 hydrogen bonds (average bond length 3.13±0.27Å); the thiolation state 6MG0 has 9 bonds (average length 3.09±0.24Å); and the condensation state 6MFZ has only 6 bonds (average length 3.13±0.27Å).

These changes observed in hydrogen bonds during confirmation changes prompted us to examine whether these interactions are functionally relevant. One prediction is that, besides the highly conserved glycine, the chemical properties of other residues involved in these interactions should also be conserved. Therefore, we turned to the 1,161 MiBiG C+A+T alignment to check the chemical conservation of the residues interacting with the G-motif (S5 Table). The cross-species conservation of chemical properties supported our prediction (Table 1). Of note, among these residues, only T491 and N479 use the hydroxyl or amide group in their side chains to form hydrogen bonds. In contrast, the other hydrogen bonds are formed by the commonly occurring α-carboxyl group and α-amino group in the main chains. The high chemical conservation among these related residues suggests a strong selective pressure to maintain desired functions.

The small size and lack of a side chain of glycine make it unique among proteinogenic amino acids. We observed that two amino acids, N577 and F493 in LgrA, locate near the glycine in the G-motif (G505). One possible explanation for the high conservation of glycine in the G-motif is that its small size provides greater structural flexibility, reducing the likelihood of collisions with neighboring residues. To check this hypothesis in known structures of A domains, we analyzed all available structures of AMP-binding-domain-containing proteins from the PDB database (39 sequences existing in a total of 95 structures, including different conformations or ligands, S6 Table). Of these 39 sequences, 30 are from NRPS pathways (selecting 20 different substrates), 6 from NRPS-like pathways (selecting 5 different substrates, with the substrates of the carboxylate reductase all being considered carboxylate), and 3 are from D-Ala-ligases (DltA) pathways (selecting substrate D-Ala). Notably, NRPS-like carboxylate reductases (CARs) do not contain the G-motif, showing a distinct evolution path despite also containing an AMP-binding domain (S6 Table, "overview" sheet). Besides CARs, the G-

**Table 1. Representative chemically conserved residues in or interacting with the G-motif.**

| The position of conserved residues | | Amino acid composition | Properties of amino acid residues (side chains) |
|---|---|---|---|
| In the LgrA protein | In MiBiG C+A+T module MSA* | In MiBiG C+A+T module MSA | |
| T491 | T1927 | T:61.67% S:15.16% | Polar, -OH; the donor and acceptor of hydrogen bonds |
| I504 | I1959 | I:88.29% V:5.25% L:3.45% | Hydrophobic |
| G505 | G1960 | G:99.66% | |
| K506 | R1961 | R:46.25% K:23.84% | Positively charged; the donor of hydrogen bonds |
| I578 | L2094 | L:60.9% I:29.03% V:7.67% | Hydrophobic |

Conserved residues in or interacting with the G-motif have significant chemical properties. * This first letter is the prevalent amino acid in this position of MSA.

motif is conserved in all 30 NRPS A domains, and 3 NRPS-like A domains (S22 Fig). However, except for the NRPS-like protein poly-ε-Lys synthetase (Pls-A, PDB: 7WEW), all AMP-binding-domain-containing proteins have equivalent N577 and F493 in LgrA, even for those without a G-motif (S23 Fig). The equivalent N577 and F493 have side chain sizes (S5 Table). In these known structures, the G-motif is near the adenylate part of the ligand (S23 Fig), also suggesting a potential gatekeeper role of the G-motif.

By simulated mutagenesis using PyMOL, we found that the position of glycine in the G-motif only permits small amino acids such as glycine and alanine. In simulations, mutating this residue into amino acids with larger volume or inflexible loops resulted in a disturbance of the residue N397 and S491 in Fig 4A and 4B (equivalent F493 and N577 in LgrA, Fig 3B–3D).

To further verify the importance of this highly conserved glycine, we performed mutation experiments on the "G-motif" in a monomodular NRPS FmqC involved in the biosynthesis of fumiquinazoline C (FQC) in *Aspergillus fumigatus* (Fig 4C). In this system, a tripeptide precursor fumiquinazoline F (FQF) was first synthesized by FmqA, and then the indole side chain of FQF oxidized by the FmqB. Then, the monomodular NRPS FmqC, activated L-Ala to create the product of fumiquinazoline A (FQA), thereby constructing a new $C_2$–N bond and finally forming FQC [93] (Fig 4D). The *fmqC* gene deletion resulted in the disappearance of FQC as the pathway diverged towards a putative compound (termed compound **1**), as indicated in the first two rows of S4 Fig. This is consistent with a previous study in *A. fumtigatus* AF293 [94]. We reintroduced *fmqC* into the original locus in the Δ*fmqC* background and generated one control strain with a full-length copy of *fmqC* in *A. fumtigatus*. The control strain could produce FQC just as the wild type (S24 Fig). Then, the glycine in the G-motif of FmqC (G409) was mutated to six other representative amino acids (A, alanine, R, arginine, D, aspartic acid, P, proline, W, tryptophan and Y, tyrosine), to generate six FmqC (G409) mutants in *A. fumtigatus*, respectively. These mutations cover all combinations of structural effects estimated by Missense3D [95] (S7 Table).

Following LC-MS analysis, the FQC yield of all mutants is lower than the control. While the G-motif mutants G409A or G409R didn't strongly reduce the yield of FQC, the other four mutants substantially reduced the yield of FQC (Fig 4E and 4F). For example, when the glycine was mutated into proline (G409P), the yield of FQC was only around 3% of the control (Fig 4F and Table A in S1 Text). Although we observed no accumulation of the precursor FQF in strains with substantially decreased FQC, we detected compound **1** as the divergent product in all these strains instead (Fig 4D–4F). Our mutation experiment is consistent with the potentially important roles of the conserved glycine in the G-motif.

**2.2. Conserved motif at the start of the A domain.** A conserved alanine located at the start of the Pfam annotated A domain was also identified through standardized multiple sequence alignment (Fig 3A, the first sequence logo in the first panel). This residue is

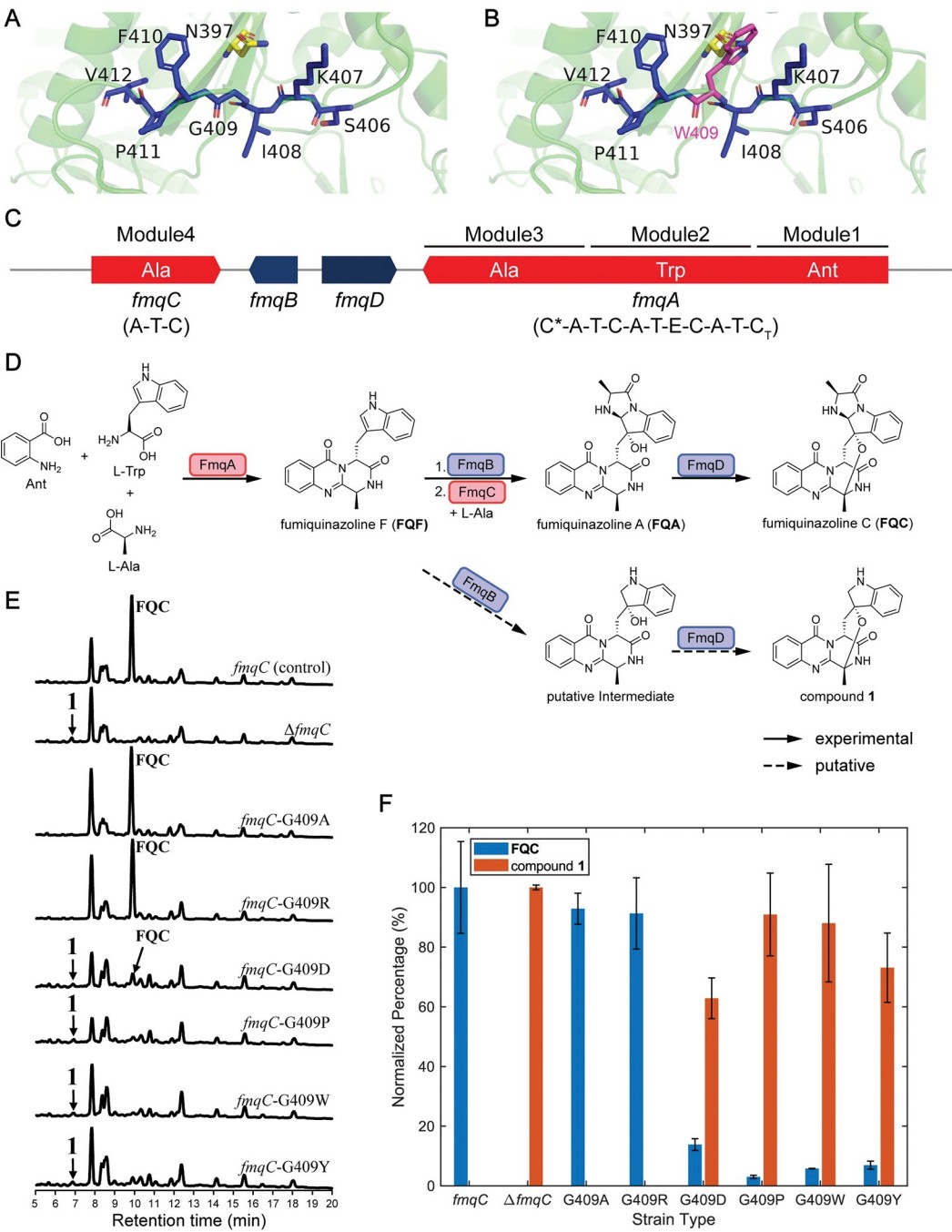

**Fig 4. Mutations of G-motif G409 in FmqC support the importance of the conserved domain in the biosynthesis of fumiquinazoline C. A.** Residues in and near the G-motif in the predicted structure of FmqC by Phyre2 [96]. G409 is the conserved glycine in the G-motif. Residues in G-motif are marked by blue. The residues N397 and S491 (equivalent to F493 and N577 in LgrA, Fig 3B–3D), which may collide with G409 are marked by yellow and cyan. **B.** Same as that in A, but with simulated mutation of G409W. The mutated tryptophan is marked in magenta. **C.** The *fmq* gene cluster responsible for the production of FQC. Two NRPSs, *fmqA* and *fmqC*, are filled in red with their substrate selectivity marked. Ant: non-proteinogenic amino acid anthranilate. C* represents a truncated and presumably inactive C domain. $C_T$ represents a terminal condensation-like domain that catalyzes macrocyclization reaction. **D.** The biosynthetic pathway for FQC is depicted, along with how it diverges into the production of compound **1** in the absence of functional FmqC. **E.** LC-MS analysis of the control *fmqC* (first row), Δ*fmqC* (second row), and six point mutation strains (3rd to 8th row). **F.** Normalized yield of FQC and compound **1** in different strains. For FQC, the yield is normalized by its production in the wild-type strain. For compound **1**, the yield is normalized by its production in the *fmqC* gene deletion strain. Error bars show standard deviations.

positioned 11(±0.4) aa upstream of the A1 motif, and 124(±4) aa downstream of the C7 motif. This residue is found at A221 in the LgrA sequence. By analyzing hydrogen bonds based on the LgrA structures, we noticed that this conserved A221, V222, and another relatively conserved proline (P217 in LgrA), interact with residues in the A1 motif (L229, T230, Y231, and K232, where the A1 motif spans 229–234 in LgrA with sequence LTYKQL), as well as the back side of the A4 motif (L412 and I414; where the A4 motif spans 395–398 in LgrA with sequence FDGS). Direct interactions were also observed between A410, L406, and Y231 (S25 and S26 Figs). The chemical characteristics of residues anticipated to interact with the conserved alanine are also conserved in the alignment of 1,161 sequences, similar to the G-motif (S5 Table). We termed this motif "Aα1", marking the start of the A domain. Previous studies have reported Aα1 as a natural recombination site in Myxobacteria [97] and applied it in synthetic biology for diverse NRPS products [98].

**2.3. Conserved motif at the first helix of the T domain.**   The other two conserved residues distant from known motifs are glutamic acid and leucine, both found at the start of the Pfam-anropnotated T domain (Fig 3A, the third sequence logo in the first panel). They were found at E700 and L711 in the LgrA sequence. The conserved glutamic acid is 10(±0.04) aa from the conserved leucine, which is 8(±0.6) aa before the first residue of the T motif. The T domain is known to be a distorted four-helix bundle with an extended loop between the first two helices [99]. According to the LgrA structure, the E700 and L711 are located at the N and C terminus of the first helix of the T domain, respectively. Distinct from the G-motif and the conserved A221, we found no hydrogen bonds outside the interior of this alpha helix. Instead, the E700, L711, and other conserved aliphatic and aromatic amino acids between E700 and L711, take part in the formation of the hydrophobic core (S27 Fig), agreeable with previous reports that the four helices were bound together by the hydrophobic interaction [100]. We termed this motif "Tα1", marking the start of the T domain.

## 3. SCA reveals overlapped co-evolving sectors across the C+A+T module

Coevolution analysis relies on proper sequence alignment, standardized NRPS structures can facilitate. After examining the sequence properties of the standardized C+A+T module, we employed coevolution analysis to study the coupling between positions. Statistical Coupling Analysis (SCA), a commonly used technique for detecting "evolutionary units", was applied to the 1,161 aligned and standardized C+A+T sequences from MiBiG. In the SCA result, only a few top modes (representing the collective covariations of a set of residues known as "sectors") have eigenvalues (measures of the covariation magnitude) that are significantly larger than those obtained by random shuffling (26 out of 2560 total, S28 Fig). The dominant first mode of SCA displays global correlations, commonly regarded as the phylogenetic association and generally ignored in the subsequent analysis [55]. The remaining sectors, sorted in descending order based on eigenvalues, were referred to by their order and sign of the eigenvalues (see Method for details). Sectors II(+) and II(-) immediately after the phylogenetic mode contain 122 positions in the MSA with significant patterns of correlation (Fig 5A, two groups labeled green and magenta for II(+) and II(-), respectively). The II(+) and II(-) sectors are orthogonal because they are the opposite directions of the same eigenvector. The II(+) sector primarily contains residues in the C domain, while the II(-) sector spans the entire NRPS module but does not contain any residues located in the pocket region of the A domain (Fig 5A).

Then we searched for sectors with a high proportion of residues from the A domain pocket region. The sector densely covering the A domain pocket region appears in the sixth biggest mode (Fig 5A, labeled as IV(-) sector with red dots), which includes three residues in the specificity-conferring code [29]. While the IV(-) sector varies nearly independently from the II(-)

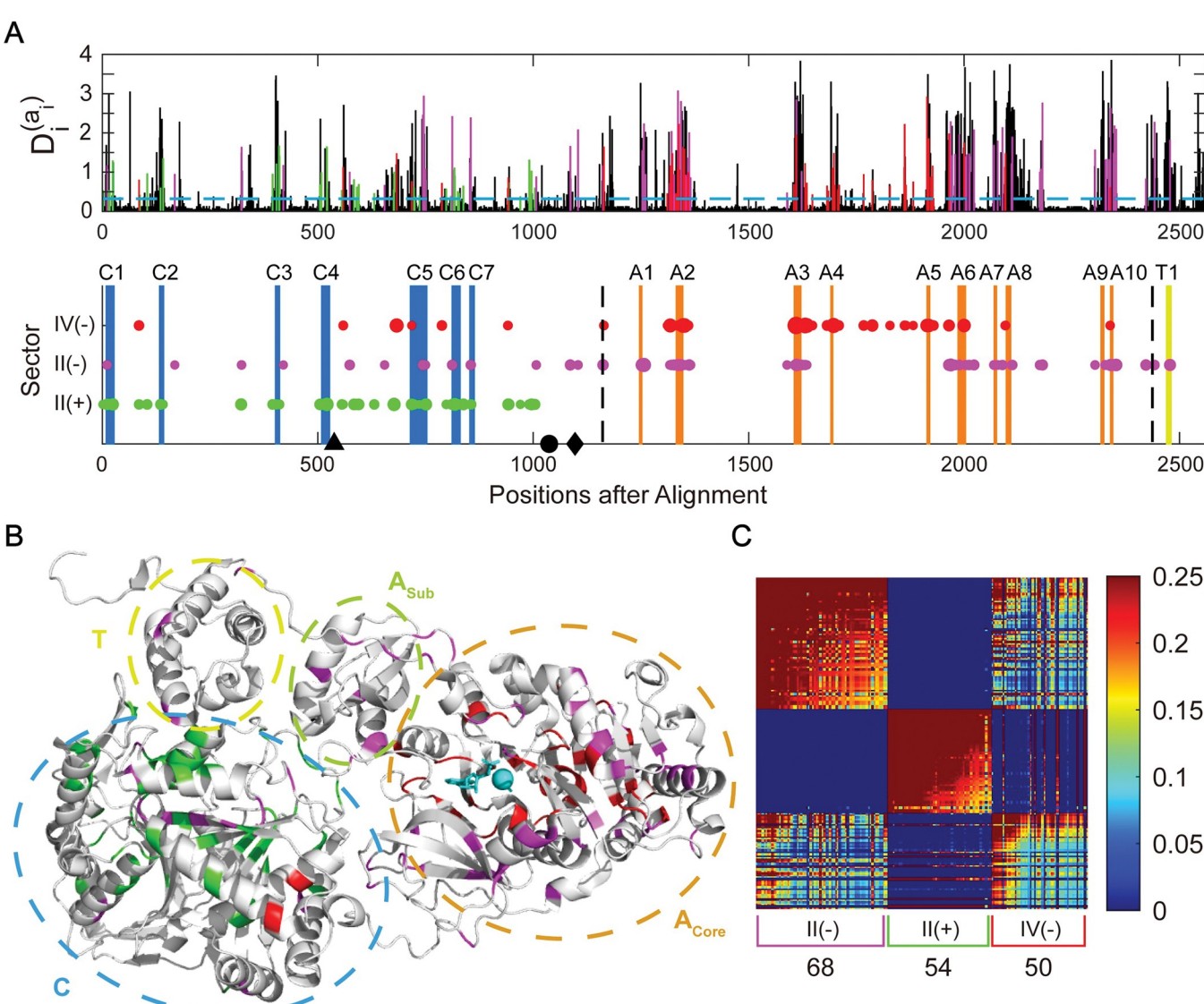

**Fig 5. Statistical coupling analysis reveals overlapped sectors across the C+A+T module. A.** The upper panel shows the conservation of residues in a multiple sequence alignment of 1,161 NRPS modules (containing the C, A, and T domains), quantified by the relative entropy $D_i^{(a_i)}$ in SCA method. The mean conservation level (0.32) is marked by the blue dashed line. In the lower panel, there are three groups of positions (II(+) with green, II(-) with magenta, and IV (-) with red, termed "sectors". Their corresponding conservations are marked in the same color in the upper panel. Blue bars mark C domain motifs from C1 to C7. Orange bars mark A domain motifs from A1 to A10. Yellow bar marks the T domain motif T1. Domain boundaries annotated by Pfam are divided by vertical black dashed lines. Black triangle marks the re-engineering point in the C domain reported by Bozhüyük et al. [27], black circle marks the re-engineering point in the C-A inter-domain reported by Calcott et al. [28] and black diamond marks the re-engineering point in the C-A inter-domain reported by Bozhüyük et al. [26]. **B.** Mapping three groups of correlated conservation positions into the three-dimensional structure of the NRPS module (PDB 4ZXI, containing the C, A, T, and TE domains. TE domain is hidden for clarity). Three sectors are marked in the same color as that in **(A)**. C domain, A core domain, A sub domain, T domain are circled by blue, orange, yellow green, and yellow dotted line, respectively. Gly and AMP are substrates of this A domain. They and Mg$^{2+}$ (for catalysis) are colored cyan. **C.** Heatmap of the SCA matrix after reduction of statistical noise and of global coherent correlations (see Method for details). Each sector is marked by the corresponding color bracket under the heatmap, with the number of contained residues listed. 68, 54, and 50 positions belong to the II(-), II(+), and IV(-) sectors, respectively. In each sector, residues are ordered by descending contributions, showing that sector positions comprise a hierarchy of correlation strengths.

sector, residues contributing to the IV(-)and the II(-) sectors share a significant degree of covariation (Fig 5C), suggesting entanglements between substrate-specifying residues in the A domain and other regions of NRPSs. In detail, there are 5 positions (residue 85, 559, 718, 941,

942 in the MSA of Fig 5) shared between the II(+) and IV(-) sectors, and 6 positions (1164, 1351, 1358, 1605, 1631, 1635, 1965) shared between the II(-) and IV(-) sectors. In the II(+) sector, most positions are highly correlated. Therefore, overlapped positions with strong correlations were observed between the II(+) and IV(-) sectors (Fig 5C). We noted that some residues of the C4-C5 and C5-C6 intermotifs are located in the II(+) and IV(-) sectors. The N-terminal of the C5-C6 intermotif is adjacent to the G-A6 intermotif region in the crystal structure, which is part of the A domain substrate binding pocket (A3-A6, S29 Fig). Therefore, there is a possible coevolution in the overlapped region between the II(+) and IV(-) sectors.

Our coevolution analysis provides valuable insights for defining boundaries in NRPS re-engineering. According to previous works on re-engineering new proteins by recombination methods [23], it's recommended to avoid cutting into coevolution sectors when recombining sequences. However, our analysis indicated that there is no simple cutting point that can clearly separate all major sectors, as these sectors overlap with each other. Nevertheless, our findings are consistent with previously successful examples of re-engineering NRPSs in their cutting point for recombination. As shown in the second panel of Fig 5A by black marks, Bozhüyük et al. cut the NRPS at the beginning of the IV(-) sector in their 2019 work of the exchange unit condensation domain for NRPS re-engineering(XUC, black triangle); both Calcott et al. and Bozhüyük et al. in their 2018 work of defining the exchange unit (XU) as a functional unit to cut NRPS after the ending of II(+) sector (black dot and diamond, respectively).

In addition, we also analyzed 685 C+A+T+C sequences and 245 C+A+T+E sequences by SCA (S30A and S30B Fig). In both sets of sequences, we found consistent sectors with the C+A+T analysis (S31 Fig). In C+A+T+C, the first two sectors locate to the two C domains, respectively, and the third sector spans across the four domains. In C+A+T+E, one sector locates to the C domain, and another sector spans across the four domains. Intriguingly, the third sector spans across the T and E domains, suggesting a possible cut point located at the beginning of this sector, between the A and T domains (S30 Fig).

## 4. Factors influencing the mapping from the A domain sequence to its substrate specificity

**4.1. Entanglement between substrate specificity and phylogenetic history.** We then utilized a modified version of SCA to investigate the relationship between substrate specificity and residues in the A domain. Our analysis was based on 2,636 standardized A domain sequences with experimentally confirmed substrates, obtained from bacteria (2370 sequences), fungi (215), and other sources (51). These A domains were gathered from the supplemental material of SeMPI 2.0 research article [37] (S8 Table). These A domain sequences were aligned, and then their substrates were attached to the last column of the alignment (see Method for details). We primarily focused on modes that co-vary with the substrate column (Fig 6A). We found residues contributing to substrate-related sectors present throughout the A domain, with a higher concentration in the A3-A6 pocket region and a more scattered distribution in the A2-A3 region. Notably, the phylogenetic sector (Fig 6A, sector I) receives the second-largest contribution from the substrate column. It has long been recognized that phylogenetic relatedness results in consistent correlations across the whole sequence. As a result, substrate specificity, which is highly linked to the specificity-conferring code, is also linked to phylogenetic covariation throughout the A domain. This observation provides an explanation for the diversity of the specificity-conferring code even for the same substrate.

**4.2. Architecture of the substrate binding pocket relates to the diversity of the specificity-conferring code.** Generally, SCA method gives higher weight to conserved positions [55]. Interestingly, some substrate-related sectors tend to locate in highly variable regions (Figs 6A

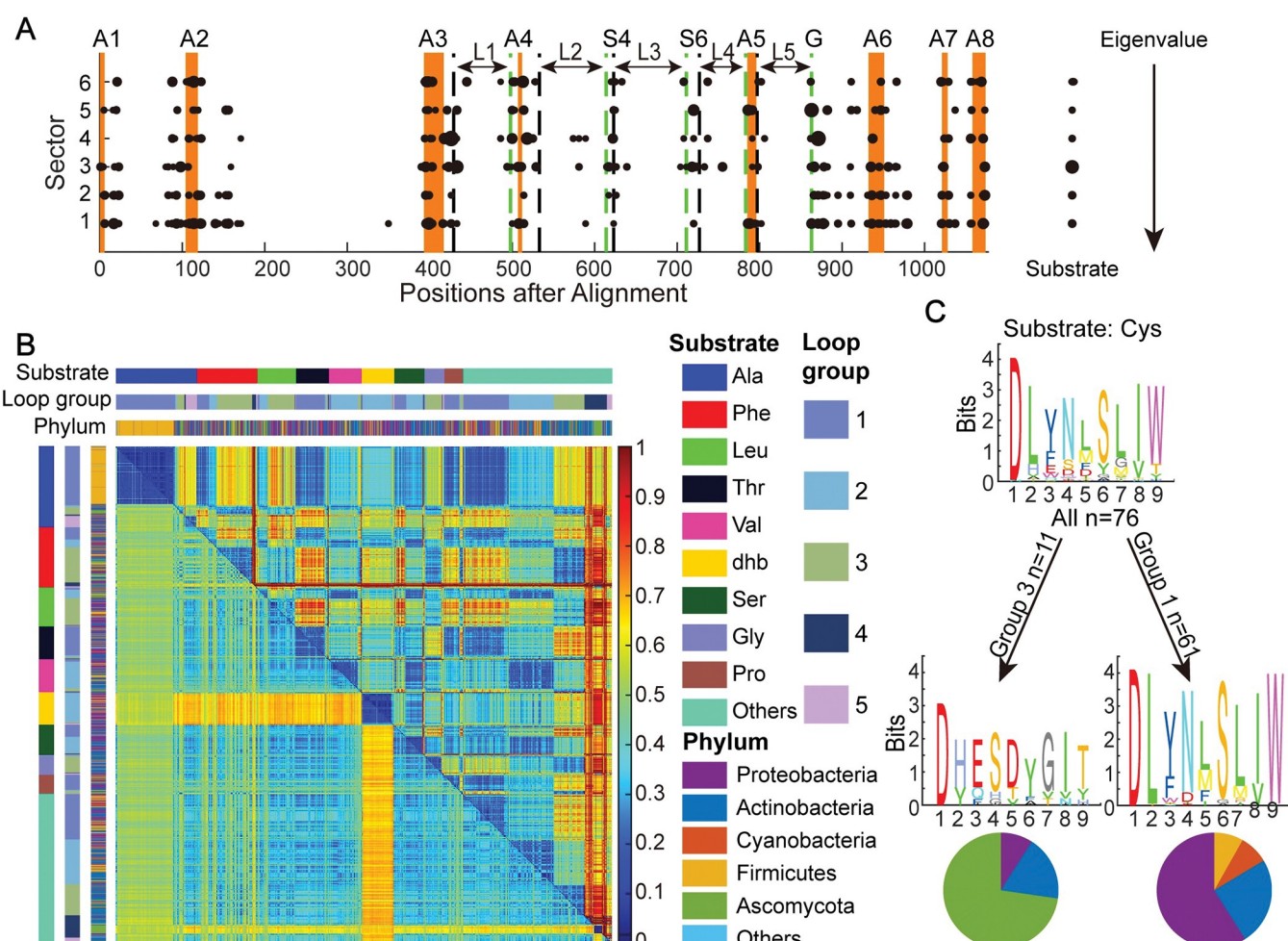

**Fig 6. The specificity-conferring code of the A domain is correlated with loop length and phylogeny. A.** SCA of 2,636 A domain sequences, together with their substrate specificities attached to the last column of the multiple sequence alignment. Six sectors with a high contribution from the substrate column (>0.05, the size of points on the left scales the substrate's contribution to the sector, see Method for details) are sorted by their eigenvalues. The size of points scales its contribution to the sector. Orange bars mark the A domain motifs from A1 to A8. The start and end of the five loop regions are marked by black and green dotted lines, respectively. S4 and S6 are the 4th and 6th of the specificity-conferring codes. G is the G-motif. **B.** Distance matrix of A domain. Upper right on the heatmap is the Euclidean distance of the loop length as a 5-element vector. Lower left on the heatmap is the sequence distance of the A domain. The matrix is sorted by the substrate specificity followed by the loop length group. Substrates, groups of loop length, and phylum of these A domains, are shown by colors in sidebars. **C.** Example showing that A domains conferring identical substrate exhibit distinct specificity-conferring codes, when they are categorized into different loop-length-groups. Phylum composition in each group is shown in the pie chart.

and S31). In aligning to PDB structures 1AMU and 4ZXI, we observed that these variable regions represent five protein loops in the A domain binding pocket (Fig 6A, loops region marked by arrows between motif A3-A6): A3-A4, A4-S4, S4-S6, S6-A5, and A5-G. $A_n$ is the *n*-th core motif of the A domain. $S_n$ is the *n*-th residue in the specificity-conferring code. G is the highly conserved glycine in the G-motif aforementioned).

Constructing an efficient MSA from highly varied regions is challenging. Therefore, to further understand the functions of these specificity-related loops, we compared their basic architectures, which are quantified by their length in amino acids. Loops' starts and ends are highly conserved motifs and residues, so loop lengths are relatively independent with MSA. These highly conserved sequence pieces help to anchor the alignment, so the length of the variable regions between motifs (i.e., the loop length) does not influence the alignment much. In the

variable regions, length discrepancies correlate with larger sequence distances. Therefore, we estimated the Euclidean distances between pairs of A domains using the lengths of the five loops as a five-element vector. We observed that this loop-based distance is connected with, but not totally determined by, the similarity of sequence alignments and the phylogeny of A domains (Fig 6B).

This loop-based distance matrix was clustered hierarchically, generating five groups based on their loop-length vectors (S32 Fig). Except for substrates Ala, dhb, and aad, one substrate typically corresponds to multiple loop-length groups. Importantly, the diversity of the specificity-conferring code is significantly reduced within each loop-length group. For example, for the substrate cysteine, the specificity-conferring code in the first loop-length group is readily distinguishable from that in the third loop-length group, despite these two groups containing A domains from distant phylogenies. Statistically, the diversity of the specificity-conferring code is reduced by identifying its loop-length vector group (S33 Fig). To visualize the reduction, we decoupled the specificity-conferring code for specific substrates along the phylum and loop group (S34–S41 Figs). In addition, we compared the loop length and loop group distribution between bacteria and fungi in MiBiG and the large dataset (S42 Fig). We found a significant difference in loop group preference: the loop group 3 is dominant in fungi (85.86%) but it prefers the loop group 1 (37.89%), 2 (28.43%), and 3 (25.10%) in bacteria.

In summary, we highlighted the importance of loop length for A domain substrate, which should be considered in the future A domain substrate prediction.

## 5. The NRPS Motif Finder online platform

Overall, such standardization enables statistical characterization of the sequence-function connections, supporting the use of known core motifs as "coordinates" of NPRS. To facilitate researchers in related fields, we constructed an online platform, "NRPS Motif Finder", for parsing the motif-and-intermotif standardized architecture of NRPS from its coding sequences (Fig 7, http://www.bdainformatics.org/page?type=NRPSMotifFinder). It takes any query amino acid sequences in FASTA format and can be interacted with by clicking on the website result page for more information. In addition to the 26 well-established motifs, the NRPS Motif Finder also supports the three new motifs proposed in this article. The NRPS Motif Finder online version is based on Python, and the result can be downloaded in Excel format for further analysis. For researchers with basic programming skills, we recommend the NRPS Motif Finder Matlab version for timely updates (the source code is available in S4 and S5 Files).

NRPS Motif Finder also supports the classification of 18 subtypes of the C domain, providing the first tool to annotate major fungal C domain subtypes (Fig 7). Furthermore, we provide all HMM files used in C domain subtype classification in S1 File for other researchers.

## Discussion

About two decades ago, the core motifs of NRPSs were mapped out based on the early crystal structures and a limited number of sequences available at the time [60]. Since then, modules, domains, and motifs have been extensively utilized by annotation algorithms like antiSMASH [35] and Pfam [38], as well as experimental investigations [36,37], albeit with differing standards. Nowadays, with the rapid expansion of microbial sequencing data, systematic characterizations of NPRS are becoming possible. In this work, we presented a motif-and-intermotif architecture of NRPSs by partitioning the C, A, T, and TE domains by well-established core motifs. Guided by prior knowledge about these 19 motifs and domain characterizations, our standardized architecture presented a "common language" for processing NRPS sequences

**Fig 7. Demonstration on the result panel of the NRPS Motif Finder.** The NRPS Motif Finder result panel provides an interactive interface. The whole result could be navigated by scrolling the page, and details about motif and intermotif could be viewed by clicking the corresponding components. Especially, the predicted subtype and confidence score are displayed for C domains. The general statistics about the NRPS architecture are displayed on the right for comparison. The results could be downloaded in table format.

from various sources, enabling us to gain statistical insights from a large number of sequences without the need for manual curation.

Evolution imprints protein functions into their sequences. With the help of such a standardized architecture, novel insights acquired from sequence statistics exhibit amazing concordance with our point mutation experiments, as well as earlier findings derived from structure or re-engineering efforts. For example, we identified novel conserved residues distant from known motifs, among which the G-motif seems the most intriguing. It centers on an exceptionally conserved glycine, where a single point mutation can abolish the enzyme's function, as described in Fig 4. Interestingly, by using visual cues from protein structure, the G-motif has been used as the cut point of A subdomain for successfully re-engineering A domains [101,102]. Crüsemann et al. first proposed the A domain re-engineering strategy by A subdomain swap guided by evolution in 2013 [101]. Subsequently, Kries et al. proposed another A subdomain swap strategy in 2015, as they discovered that the substrate binding region is a flavodoxin-like subdomain [102]. This subdomain starts from the middle of A3 and A4, and ends from the middle of A and A6 (the ending is exactly the G-motif, VPIGAPI in the

PheA). Recently, Thong et al. also used this A subdomain swapping strategy by CRISPR-Cas9 and successfully re-engineered A domain substrate specificity by subdomains from a range of NRPS enzymes of diverse bacterial origins [103].

NRPS complexes undergo substantial conformational changes as they exert multistep synthetase activity [63]. Identifying key residues associated with such structural flexibility is crucial for understanding NRPS functionality. In our analysis, three additional conserved motifs were identified. We hypothesized that the G-motif in the loop might be one of these key sites for structural flexibility, acting as a hinge when NRPS switches between distinct states. First, the number of hydrogen bonds around the conserved glycine changes substantially during the three function states. Also, glycine is the smallest of the 22 proteinogenic amino acids [104], allowing the greatest structural flexibility. Additionally, it has been reported that the G-X-Y or Y-X-G oligopeptides motif provides the flexibility necessary for enzyme conformation change for catalysis, where X and Y are polar and non-polar residues, respectively [105]. G-R/K-P in the G-motif fit this G-X-Y structure. Furthermore, several enzymes have demonstrated the important function of conserved glycine residues in structure flexibility. For example, conserved G76 contributes to active-site loop flexibility in the pepsin [106], and conserved G316 and G324 are the structural basis of efficient metal exchange in the cadmium carbonic anhydrase of marine diatoms [107]. In addition, the importance of this highly conserved glycine was supported in the fungal FQC biosynthesis system. We noted that the large amino acid in the 409 position mostly collides with N397 in simulated mutations (F493 in this position of MiBiG NRPS sequences, Fig 3B–3D). This may explain why mutating into large amino acids at the 409 position always reduces the FQC yield. Nevertheless, mutation into a small amino acid proline also strongly reduces the yield. It is possible that the proline's side chain locks the dihedral angle $\Phi$ of the protein backbone at approximately $-65°$ and causes significant conformational restriction [108]. Our results implied that the flexibility of glycine in the G-motif might be important for conformational change during A domain functions, so conformational restrictions in the G-motif impede A domain function.

We have also identified other conserved residues that may play a role in the functionality of NRPS. The conserved Ala and Pro at the start of the A domain (A$\alpha$1) may bridge the A1 and A4 motifs. Similarly, the conserved Glu acid and Leu in the first helix of the T domain (T$\alpha$1) may contribute to the proper folding of the T domain, much like the three conserved Gly residues contribute to the folding and function of the green fluorescent protein [109]. From a functional perspective, these residues take part in the formation of the hydrophobic core [99,100]. In addition, 4'-phosphopantetheine cofactor required by the T domain function near residues Y748, L751, and F752 [110] (numbered in LgrA. They are L65, L68, and F69 in the original research for holo-TycC3-PCP). Besides, Y748 and F752 also are part of the hydrophobic interface, which interacts with the C domain in docking process (in LgrA. V2534 and F2538 in original research for PCP2-C3 didomain from fuscachelin) [111]. In summary, the conserved residues we identified may contribute to the stability and flexibility of NRPS proteins in large conformational changes for product synthesis. In the future, it would be meaningful to experimentally explore these new motif candidates more thoroughly, to gain deeper insights into the conformational dynamics during NRPS functioning.

Another point of consistency between our sequence characterization and earlier re-engineering studies is the SCA sectors that we identified. Among the three sectors we analyzed, the cutting point from Bozhüyük et al.'s 2019 work [27] locates right at the beginning of the red sector, which contains residues enriched in the binding pocket of the A domain. The cutting points of Calcott et al. and Bozhüyük et al.'s 2018 work [26,28] both locate to the end of the green sector, which primarily represents covariation mode in the C domain. Congruence

between sector boundaries and successful re-engineering points substantiated SCA's potential to discern the "unit of evolution".

Nonetheless, our analysis also dissected and reemphasized the complexities in re-engineering NRPS. For example, sectors obtained from SCA intersperse with each other, even share covariation residues. Therefore, there is no universal cutting point in an NRPS module that does not disturb any of the major sectors. This observation is in line with difficulties in re-engineering NRPS [23]. The success of Bozhüyük et al. and Calcott et al. [26–28] may be partially attributed to their cutting points being right beside some of the major sectors, and they utilized sequences from NRPSs relatively close on phylogeny. It may be possible in the future to search phyla for NRPS systems with non-overlapping covariation sectors and use these systems as "building blocks" that can be recombined with greater freedom. As homologous recombination requires conserved sequence, the standardized architecture could aid in establishing the appropriate "cutting points" in such potential NRPS systems for re-engineering.

The phylogeny's entanglement with the substrate-specific binding pocket adds another layer of complexity in manipulating NPRSs. The contribution of the phylogeny sector to substrate specificity suggests non-degeneracy of the sequence space in determining substrate: the same substrate can be selected by distinctive binding-pocket sequences, which drift with phylogenies and may recombine with each other. Such non-degeneracy leads to the diversity of the specificity-conferring code, causing troubles in the targeted design of the A domain specificity. We demonstrated that knowing the length of the five loops in the pocket region reduces the specificity-conferring code diversity, and tried to provide an explanation by a causal diagram (S43 Fig, the causal diagram is a method for causal inference [112]). It suggests a connection between the specificity-conferring code and loop length group for a specific substrate (S34–S41 Figs). However, a mechanistic understanding of substrate selectivity is still needed to guide the rational design of the A domain. The architectural distinctions and loop length group preference between fungal and bacterial A domains discussed may shed light on the detailed mechanism of substrate selection, and help to establish the framework for developing substrate prediction algorithms that are less reliant on experimentally validated substrates.

The standardized motif-intermotif architecture not only enables efficient analysis of large datasets, but also provides a useful framework for in-depth examination of individual NRPS pathways. To facilitate researchers in related fields, we presented the NRPS Motif Finder online platform with a user-friendly tool to construct the motif-and-intermotif architecture of NRPS. The resulting architecture can be used in A domain substrate prediction based on phylogenies of A3-A6 or A4-A5. Also, this tool can be used in guiding re-engineering and new NRPS discovery, as G-motif and Aα1 both coincide with known cutting points [97,98,102]. In addition, NRPS Motif Finder supports the classification of C domain subtypes into 18 kinds, offering the first tool to cover most of the fungal C domain subtypes.

In terms of limitations, most of our results are based on computation. Despite the high degree of concordance between computational predictions and our point mutation experiments on the G-motif, more comprehensive experiments might be developed in the future to explore the roles of other conserved residues we identified. Additionally, given the diversity of NRPSs, despite our use of the largest A domain database with known substrates, A domains selecting rare substrates may be unrepresented. Moreover, the identification of the new C domain subtypes, while promoting the understanding of fungal NRPS, also highlighted the current limitation of the dataset, which is heavily biased towards bacterial BGCs. The catalytic mechanism and evolutionary history of new C domain subtypes are still unclear and require more investigation. Overall, our effort is a preliminary investigation into the possibilities of standardized architecture in modular enzymes. Much more discoveries could be achieved in the future with the rapid expansion of microbial sequencing data.

## Method

### Data acquisition

We downloaded the MiBiG database version 2.0 [33], then extracted sequence information from this database. In this research, only C/A/T/E/TE domains from NRPS clusters were considered. Finally, 326 NPRS sequences from 264 species were obtained, with 1,864 A domains, 1,765 C domains, 1,803 T domains, 310 E domains, 280 TE domains. In total, there are 1,161 C+A+T modules, 685 C+A+T+C modules, and 245 C+A+T+E modules.

A domains sequences with known substrates were gathered from the supplemental material of SeMPI 2.0 research article [37]. Some of the A domains from SeMPI overlap with those in MiBiG (http://sempi.pharmazie.uni-freiburg.de/database/MiBiG).Sequences which had deleted in MiBiG database or UniProt [113] were removed from our dataset. Finally, we collected 2,636 A domain sequences. Sequence details see S8 Table. Pfam seed alignments were downloaded for comparison with antiSMASH domains (PF00668 for C domain, PF00501 for A domain, PF00550 for T domain).

Sequences with high identity were removed (S44 Fig). All analysis was applied to the remaining sequences.

For the large dataset to confirm our results, we downloaded all complete bacterial genomes (30,984) and all assembly-level (including "contig", "scaffold", "chromosome", and "complete" levels) fungal genomes (3,672) from NCBI (as of 2022/10/23). Genomes were deduplicated by Mash distance with a cutoff of 0.004 ($>$ = 99.6% genome similarity) using Mash tool v2.3 [114]. Representative genomes were chosen by picking the genome with the longest genome size. After removing redundancy, we had 16,820 bacterial and 2,505 fungal genomes. The information of used genomes was summarized in S9 Table. Species information was obtained by TaxonKit [115].

### Length threshold for multi-domain NRPS from MiBiG

Considering the NRPS sequence length distribution, we filtered sequences by the following thresholds. For C+A+T, the sequence length should be more than 1000 and less than 1300; for C+A+T+C, the sequence length should be more than 1500 and less than 1750; for C+A+T+E, the sequence length should be more than 1350 and less than 2350. In practice, 14 sequences in the C+A+T+C type are removed because their secondary C domain, rather than the first C domain, align with the first C domain of other sequences. And a few sequences from C+A+T +E were removed because they are incomplete at start.

### Multiple sequence alignment

If not specifically stated, the MSA in our work were constructed by Clustal Omega 1.2.4 [116]. For the construction of C domain subtype reference HMM files, we prepared MSA by Muscle v5 referring to previous studies [5,61].

### antiSMASH annotation

Data obtained from MiBiG database has already been annotated by antiSMASH 5.1 [33]. BGCs in the large dataset were annotated by antiSMASH 6 [35].

### Conserved motif detection for NRPS domain

As shown by S45 Fig, first, manual curation of the conserved motifs from the literature was performed for all domains. That resulted in 7 motifs for C, 10 motifs for A, 1 motif for T, 7

motifs for E, and 1 motif for TE [60]. A table of these conserved motifs according to the previous literature was summarized in S10 Table.

Second, we obtained reference sequences for each type of NRPS domain. The reference sequences of the A, T, and TE domains were from surfactin A synthetase C which has resolved crystal structure (PDB: 2VSQ) [58]. The reference sequence of the E domain was retrieved from the initiation module of tyrocidine synthetase A (PDB: 2XHG) [59]. The reference sequences of C domain are described in the section of C domain subtypes classification. The motif locations on all reference sequences above were manually identified, according to the results from step 1.

Third, for every query sequence, global alignment was performed against possible reference sequences. Regions on the query sequence that align with the conserved motifs in the reference sequence were recorded as "motifs".

For clarification, only sequences with the prevalent motif length were used in the construction of motif sequence logos. The prevalent motif lengths of each domain were recorded in S10 Table.

## C domain subtypes classification

First, we curated sequences from the E domain and 18 subtypes of the C domains as references. 14 of 18 subtypes of the C domains were sourced from the CDD [69], and the remaining 4 were obtained from literatures [33,66,70,71] (S1 Table). The original "CT" subtype sequences were classified into CT, CT-DCL and CT-A subtypes guided by our phylogenetic anlaysis.

We prepared MSA for each subtype by Muscle v5 [5] and constructed reference HMM files by hmmer v3.1b2 [74]. The MSA results of each C domain subtype were attached in the S2 File.

Finally, we aligned the query sequence to profile using hidden Markov model alignment. In the NRPS Motif Finder online version (python), it is achieved by the hmmscan function in hmmer3. In the NRPS Motif Finder Matlab version, it is achieved by the hmmprofalign function in Matlab.

## Calculation of mutual information

In generating the fourth panel of Fig 1A, mutual information between the residues and the chirality subtypes of C domains was calculated. The C domain subtype was defined by the antiSMASH annotation, including "LCL", "DCL", "Dual", "Starter", and "Heterocyclization". In the multi-alignment of C domains, the reduced Shannon entropy on the amino acid composition at position $i$ given the subtype, was calculated as the mutual information about chirality at position $i$.

The fourth panel of Fig 1B was obtained by calculating the mutual information between substrate specificity and amino acid composition at each position of the multi-alignment of A domains.

## Phylogenetic analysis

Phylogenetic trees of the condensation domain superfamily were reconstructed in Fig 2A. Details of sequences see **C domain subtypes classification**. Multiple sequence alignments of protein sequences were prepared with Muscle v5 with default parameter (muscle -super5) and trimmed sequence at both ends manually. The maximum-likelihood phylogenetic tree was reconstructed in IQ-TREE 2.1.2 [117], using the best-fit model of protein evolution for each alignment (LG+F+R10 model) as chosen by ModelFinder (Bayesian Information Criterion) [118]. Branch support was assessed by bootstrapping (1000 bootstrap replicates). Following

previous research, papA and WES are taken as outgroups to root the phylogenetic tree [65]. See details in S3 File. The phylogenetic tree was visualized with FigTree (http://tree.bio.ed.ac. uk/software/figtree/).

### Protein structure modeling

The protein structure of FmqC was predicted by Phyre2 [96]. Phyre2 used high-score PDB templates as 5U89, 6MFZ, and 6N8E for predicting the FmqC structure, with alignment coverages of 93%, 94%, 87%, and identities of 25%, 23%, 21%, respectively. Mutation of FmqC G-motif from Gly to Ala was simulated by mutagenesis function in the PyMOL (http://www. pymol.org/pymol).

### The position of motif boundaries for multi-sequence

The motif boundaries in Figs 3, 5, 6, S30, and S31 show the average motif boundary positions from each single sequence. Motif boundary positions are consistent in different sequence after MSA other than the C domain, where the conserved motifs vary by subtypes (S10 Table).

### The calculation of conservation in NRPS protein

For a quantitative perspective, we use ConSurf [119] and AACon [120] to calculate the conservation of specific positions in the NRPS protein. The results of two methods are similar in very conserved positions, but a little different in middle conserved positions. The results are displayed in S5 Table.

### Statistical coupling analysis

The statistical coupling analysis was performed with the binary approximation method [55]. The cleaned correlation matrix is obtained using the method described in the same article [55].

To investigate substrate-related positions, we attached substrates as the last column of the multiple sequence alignment. Then we applied SCA to this modified MSA. The least frequent substrate was taken as background of this position.

### Sector identification in SCA

We use the same notation in Halabi et al. [55]. Except two sectors of the second mode with opposite eigenvalues, third sector was chosen empirically. The bra-ket notation is such that $|k\rangle$ denotes the $k^{th}$ eigenvector and $\langle i|k\rangle$ the weight for position $i$ along eigenvector $k$. The threshold $\varepsilon$ to separate significant weights along an eigenvector from statistical noise is 0.05, which was used in Halabi et al. [55].

For C+A+T, green sector is defined as $\langle i|2\rangle > \varepsilon$; magenta sector is defined as $\langle i|2\rangle < \varepsilon$; red sector is defined as $\langle i|6\rangle < \varepsilon$.

For C+A+T+C, green sector is defined as $\langle i|2\rangle > \varepsilon$; magenta sector is defined as $\langle i|4\rangle > \varepsilon$; red sector is defined as $\langle i|2\rangle < \varepsilon$.

For C+A+T+E, green sector is defined as $\langle i|2\rangle < \varepsilon$; magenta sector is defined as $\langle i|3\rangle > \varepsilon$; red sector is defined as $\langle i|2\rangle > \varepsilon$.

For A domain, from bottom to top, sectors are defined as $\langle i|1\rangle > \varepsilon$, $\langle i|2\rangle > \varepsilon$, $\langle i|3\rangle > \varepsilon$, $\langle i|4\rangle > \varepsilon$, $\langle i|5\rangle > \varepsilon$ and $\langle i|6\rangle < \varepsilon$.

## Loop length clustering

The sum of the Euclidean distance of five loops between different A domain sequences was calculated for distance matrix. To avoid the influence of local maximum value on the heatmap, the upper-bound in distance was set to 12. Then hierarchical clustering analysis is applied to this distance matrix (S32 Fig).

## Loop length profile calculation method

$$\bar{L}_{i,j} = \frac{L_{i,j} - \min(L_i) + 1}{mean(L_i) - \min(L_i) + 1}$$

Related to S32 Fig, $\bar{L}_{i,j}$ is normalized length of loop i in $j^{th}$ row. $L_{i,j}$ is actual length of loop i in $j^{th}$ row. $L_i$ is a column vector containing all length of loop i.

## Experimental Materials and General Methods

The plasmids and strains utilized in this work are listed in Table B in S1 Text. The oligonucleotide primers synthesized by Shanghai Sango Biotech are given in Table C in S1 Text. PCR reactions were carried out with FastPfu high-fidelity DNA polymerase (Transgene Biotech). *Escherichia coli* strain DH5α was propagated in LB medium with appropriate antibiotics for plasmid DNA, and plasmid DNA was prepared using the Plasmid Mini Kit I (Omega). Automated DNA sequencing was performed by Shanghai Sango Biotech. PCR screening for transformants was carried out using 2×GS Taq pcr mix (Genesand).

   *A. fumigatus* strains were cultivated in glucose minimal medium (GMM) [121] culture medium at 28°C for 3 days in the dark. The media and mycelia were extracted with ethyl acetate, and then evaporated under reduced pressure. The extract was dissolved in 1 mL methanol, and 5 μL of the solution was directly injected for LC-MS analysis. LC-MS analysis was performed on an Agilent HPLC 1200 series system equipped with a single quadrupole mass selective detector, an electrospray ionization (ESI) and an Agilent 1100LC MSD model G1946D mass spectrometer by using a Venusil XBP C18 column (3.0 x 50 mm, 3 μm, Bonna-Agela Technologies, China). Data collected in positive mode. Water with 0.1% (v/v) formic acid and acetonitrile were used as mobile phase with a flow rate of 0.5 mL/min. For analysis of the extracts, a linear gradient of acetonitrile in water (10–45%, v/v) in 30 min was used and washed with 100% (v/v) acetonitrile for 5 min.

## Gene cloning, plasmid construction, and genetic manipulations

*(a) Creation of fmqC deletion strains in A. fumigatus.* For site-directed mutagenesis of G-motif in *A. fumigatus*, *fmqC* was fully deleted in Cea17.2 to increase the rate of homologous recombination and reduce the difficulty of screening transformants. The FmqC (accession: EDP49773.1) deletion strain was created in Cea17.2 by replacing the *fmqC* with hygromycin phosphotransferase gene (*hph*) using modified double joint PCR described previously [122] consisting of the following: 1 kb DNA fragment upstream of *fmqC*, a 1.9 kb DNA fragment of *hph*, and a 1 kb DNA fragment downstream of *fmqC*. Third round PCR product were purified for protoplast transformation (fragment concentration greater than 300ng/μL). Polyethylene glycol (PEG) based transformation of *A. fumigatus* was done as previously described [123]. The fragment concentration was 3 μg per 100 μL of protoplasts. The mutants were confirmed by using diagnostic PCR and the correct transformants were used for subsequent analysis.

   *(b) Creation of fmqC mutation strain in A. fumigatus.* Seven plasmids were generated, one which included a full-length copy of *fmqC* (pYJY25), and the others were mutated *fmqC*

(pYJY26-31). The pYJY25 plasmid was assembled by amplifying 1 kb DNA fragment upstream of *fmqC*, *fmqC*, *A. fumigatus pyrG* as the selectable marker, and a 1 kb DNA fragment downstream of *fmqC*. The fragments were assembled into a full plasmid by homologous recombination using MultiS One Step Cloning Kit (Vazyme). pYJY26-31 were assembled using the same fragments and method, except that the targeted mutations of 409Gly in G-motif were introduced by using primers containing the mutation and amplifying *fmqC* in two separate PCR reactions. After plasmid construction, the fused 8kb PCR fragments with the point mutation were purified and used for transformation to TYJY81. Transformants were confirmed by sequencings to obtain TYJY82-88 for the subsequent experiments.

## Supporting information

**S1 Fig. Sequence logo of the seven C motifs among the multialignment of 1758 C domains (first row), and among six subtypes of C domains (second row) in MiBiG.** The ranges of y-axis in sequence logo figures all are 0~4.4 bits. The numbers of each C domain subtypes are 809 (LCL), 385 (DCL), 114 (Starter), 300 (Dual), 114 (Cgly), and 36 (Heterocyclization).
(PNG)

**S2 Fig. Sequence logo of the ten A motifs among the multialignment of 1859 A domains in MiBiG.** The y-axis ranges in sequence logo figures all are 0~4.4 bits.
(PNG)

**S3 Fig. Sequence logo of T1 motif and the length distribution of the T1-C1 region in MiBiG. A.** Sequence logo of the T1 motif. The y-axis range in sequence logo figure is 0~4.4 bits. **B.** Length distribution of the T1-C1 region for different subtypes of C.
(PNG)

**S4 Fig. Sequence logo of the seven E motifs among the multialignment of 310 E domains in MiBiG.** The y-axis ranges in sequence logo figures all are 0~4.4 bits.
(PNG)

**S5 Fig. Sequence logo of the TE1 motifs among the multialignment of 280 TE domains in MiBiG.** The y-axis range in sequence logo figure is 0~4.4 bits.
(PNG)

**S6 Fig. Length distributions of C, A, and T domains, in MIBIG and in Pfam seeds.**
(PNG)

**S7 Fig. Sequence logo of the twelve A motifs and two T motifs among 95,582 A domains and 86,688 T domains in bacteria.** The y-axis ranges in sequence logo figures all are 0~4.4 bits.
(PNG)

**S8 Fig. Sequence logo of the seven E motifs among 14,502 E domains in bacteria.** The y-axis ranges in sequence logo figures all are 0~4.4 bits.
(PNG)

**S9 Fig. Sequence logo of the TE1 motifs among 23,590 TE domains in bacteria.** The y-axis range in the sequence logo figure is 0~4.4 bits.
(PNG)

**S10 Fig. Sequence logo of the twelve A motifs and two T motifs among 40,458 A domains and 26,651 T domains in fungi.** The y-axis ranges in sequence logo figures all are 0~4.4 bits.
(PNG)

**S11 Fig. Sequence logo of the seven E motifs among 3,982 E domains in fungi.** The y-axis ranges in sequence logo figures all are 0~4.4 bits.
(PNG)

**S12 Fig. Sequence logo of the TE1 motifs among 4,008 TE domains in fungi.** The y-axis range in sequence logo figures is 0~4.4 bits.
(PNG)

**S13 Fig. Comparison of NRPS A and T domain architecture between bacteria and fungi source.** For comparison, only A domains which have the same motif length with reference A domain are used. Sequence numbers of intermotifs (A1-A2, A2-A3, A3-A4, A4-A5, A5-G motif, G motif-A6, A6-A7, A7-A8, A8-A9, A9-A10, Tα1-T1) are 75,407 in bacteria and 17,890 in fungi. Sequence numbers of A1-Tα1 intermotif (actually interdomain) in bacteria source are 69,440 while they in fungi source are 12,194 because only part of A domains are adjacent with the T domain. Sequence numbers of Tα1-T1 intermotif in bacteria are 85,755 while they in fungi are 25,069.
(PNG)

**S14 Fig. Comparison of NRPS E domain architecture between bacteria and fungi source.** For comparison, only A domains which have same motif length with reference A domain are used. Sequence numbers of intermotifs (E1-E2, E2-E3, E3-E4, E4-E5, E5-E6, E6-E7) are 12,875 in bacteria source and 2,852 in fungi source. Sequence numbers of intermotifs (actually inter-domain) in bacteria are 12,618 for T1-E1 and 8,353 for E7-C1 while they in fungi are 2,088 for T1-E1 and 2,530 for E7-C1.
(PNG)

**S15 Fig. Sequence logo of the seven C motifs among the multialignment of 2,572 C domains (first row), and among 15 subtypes of C domains (second row) in MiBiG.** There are 2,572 C domains with subtype prediction scores more than the threshold (200) and a count of domain subtype sequences of more than 3. The y-axis ranges in sequence logo figures all are 0~4.4 bits. The numbers of each C domain subtype are labeled at the end. For clarification, only motifs that have prevalent length are used in plotting. The actual total C domain number used in the figure for the specific motif is shown in the title. There are some gaps at both ends of the sequence for alignment. In motif C5, there are some interior gaps labeled in red to align with motif C5 of other subtypes.
(PNG)

**S16 Fig. Sequence logo of the seven C motifs among 77,152 C domains (first row), and among 13 subtypes of C domains (second row) in bacteria.** There are 77,152 C domains with subtype prediction scores more than the threshold (200) and a count of domain subtype sequences of more than 3. The y-axis ranges in sequence logo figures all are 0~4.4 bits, except it's 0~1 for the bL subtype and 0~2.1 for the PS subtype because these subtypes are few in the sequence number. The numbers of each C domain subtype are labeled at the end. For clarification, only motifs that have prevalent length are used in plotting. The actual total C domain number used in the figure for the specific motif is shown in the title. There are some gaps at both ends of the sequence for alignment. In motif C5, there are some interior gaps labeled in red to align with motif C5 of other subtypes.
(PNG)

**S17 Fig. Sequence logo of the seven C motifs among 34,269 C domains (first row), and among 11 subtypes of C domains (second row) in fungi.** There are 34,269 C domains with subtype prediction scores more than the threshold (200) and a count of domain subtype

sequences of more than 3. The y-axis ranges in sequence logo figures all are 0~4.4 bits. The numbers of each C domain subtype are labeled at the end. For clarification, only motifs that have prevalent length are used in plotting. The actual total C domain number used in the figure for the specific motif is shown in the title. There are some gaps at both ends of the sequence for alignment. In motif C5, there are some interior gaps labeled in red to align with motif C5 of other subtypes.
(PNG)

**S18 Fig. Comparison of NRPS C domain architecture between bacteria and fungi.** For comparison of intermotif length between different C domain subtypes and E domain, we chose the conserved positions which exist in all C domain and E domain as start and end of intermotif. T1/ACP1-C1/E1 ends before the conserved "Q" in C1 and E1 (the conserved "E" in LCL-A subtype). C1-C2 starts with the conserved "Q" in C1 (the conserved "E" in LCL-A subtype), and ends before the second conserved "R" in C2. C2-C3 starts before the second conserved "R" in C2 and ends before the conserved "D" in C3. C3-C4 starts before the conserved "D" in C3 and ends before the second conserved "Y" in C4. C4-C5 starts before the second conserved "Y" in C4 and ends before the conserved "G" in C5. C5-C6 starts before the conserved "G" in C5 and ends before the conserved "P" in C6. C6-C7 starts before the conserved "P" in C6 and ends before the conserved "F" in C7 (the conserved "F" in LCL). C7-A$\alpha$1 starts before the conserved "F" in C7 (the conserved "F" in LCL) and ends before A$\alpha$1. Sequence numbers of intermotifs (C1-C2, C2-C3, C3-C4, C4-C5, C5-C6, C6-C7) are 77,152 in bacteria and 34,269 in fungi. Sequence numbers of intermotifs (actually interdomain) in bacteria are 28,185 for T1-C1 and 33,176 for C7-A$\alpha$1 in LCL subtype C domain, 6,967 for E7-C1 and 6,752 for C7-A$\alpha$1 in DCL subtype C domain, 3,860 for T1-C1 and 4,495 for C7-A$\alpha$1 in Dual subtype C domain, 6,967 for T1-C1 and 6,752 for C7-A$\alpha$1 in starter subtype C domain and while they in fungi are 434 for T1-C1 and 616 for C7-A$\alpha$1 in LCL subtype C domain, 808 for E7-C1 and 1,931 for C7-A$\alpha$1 in DCL subtype C domain and 24 for T1-C1 and 21 for C7-A$\alpha$1 in Dual subtype C domain.S19 Fig.
(PNG)

**S19 Fig. Sequence logo of highly conserved positions near known motifs.** The black box shows known core motifs in the A domain. The black triangle shows highly conserved positions in multialignment from the 1,161 C+A+T NRPS sequences from MiBiG database.
(PNG)

**S20 Fig. The mutual information between residues in the A5-A6 and A domain substrate specificity.** Same as that in the fourth panel of Fig 1B, but this plot focuses on regions between motif A5 and motif A6. The position of conserved Gly is indicated by the black arrow.
(PNG)

**S21 Fig. The G-motif in different function states of LgrA structure.** Related to Fig 3B–3D. The view zoomed in toward the region near the G-motif emphasized by red sticks. Hydrogen bonds were shown in the blue dashed-line. Different domains marked by different colors (F: formylation domain, colored by magenta; T: thiolation domain, colored by cyan; A: adenylation domain, Acore (orange) covers A1-A8 of A domain, Asub (yellow) covers A9-A10, Apocket (yellow green) covers A3-A6 of A domain).
(PNG)

**S22 Fig. The sequence logo of G-motif in A domains with known structures and FmqC.** Non-NRPS means A domains from these proteins which aren't NRPS. Non-CAR means A domains from these proteins which aren't CAR. Non-NRPS&Non-CAR means A domains

from these proteins which aren't NRPS or CAR.
(PNG)

**S23 Fig. The equivalents of N397, G409 (G-motif), and S491 in FmqC mapped to known structures.** From the first to the 8-th figure, the known structures are CAR protein (PDB: 6OZ1), DltA (PDB: 7VHV), and NRPS (PDB: 1AMU, 2VSQ, 6IYK, 5WMM, 6OZV, and 7LY7). Residues in G-motif were marked by blue. The equivalent N397 and S491 were marked by yellow and cyan. The ligand molecule was marked in green. The PDB IDs of proteins are shown below. Only the CAR protein doesn't contain the G-motif, but it still has equivalents of N397 and S491. G-motif is in close proximity to the adenylate part of the ligand, suggesting a potential gatekeeper role.
(PNG)

**S24 Fig. LC-MS analysis in the strain construction.** Wild type (Cea17.2, first row), Δ*fmqC* (second row) and the control *fmqC* (third row).
(PNG)

**S25 Fig. Aα1 motif in different states of LgrA structure.** Similar to S21, but for the Aα1 motif. The view zoomed in towards the region near the Aα1 motif emphasized by red sticks.
(PNG)

**S26 Fig. The interaction near Aα1 motif in different states of LgrA structure.** Related to S25. Similar to Fig 3B, but shows chemical interactions and secondary structures surrounding the Aα1 motif at the substrate donation state (A), the thiolation state (B) and the condensation state (C). Of note, in these residues, only T230 and Y231 use the hydroxyl group in the side chain to form hydrogen bonds. Other hydrogen bonds, on the other hand, are formed by the common α-carboxyl group and α-amino group in the main chains.
(PNG)

**S27 Fig. Conserved residues of T domain in the condensation state.** The structure is obtained from LgrA in the condensation state (PDB: 6MFZ). Cyan color shows the T domain defined by Pfam. A small white region isn't covered by Pfam, although it is visually one part of the first helix of T domain. N-terminal and C-terminal were marked by green texts. Conserved resides were marked by red sticks with their one letter labels shown. Reside labels in the top view were hidden for visual clearness. **A.** Side view of T domain. **B.** Top view of T domain.
(PNG)

**S28 Fig. Eigenvalue spectra for the SCA matrix of C+A+T modules MSA and random MSA.** Eigenvalue spectra for the SCA matrix corresponding to the 1,161 C+A+T modules (top panel) and for 100 trials of randomizing sequences alignment (bottom panel). The randomization process scrambles the order of amino acids in each alignment column independently, which did not change amino acid frequencies at positions. The black dashed-line marked the maximum of eigenvalues from randomized alignments. This analysis shows that only a small part of the spectrum (26 out of 2560 total eigenvalues) is significant given sample size.
(PNG)

**S29 Fig. The interaction of C5-C6 with other domains in LgrA structure.** The formylation domain in the first module (F1) of LgrA is hidden for visual clearness. The colors of each domain are noted at the top of the figure. The C domain is split into the N-terminal subdomain (N-subdomain, covering C1-C4) and the C-terminal subdomain (C-subdomain, covering C5-C7), referring to previous research (PMID: 23756159). The active site histidine (the second histidine in C3 motif HHxxxD), the residues in C5-C6 intermotif interacting with T

domain or A domain, and the residues in T domain and A domain interacted by these residues are shown as stick format. Hydrogen bonds were shown in yellow dashed-line with distances by black.
(PNG)

**S30 Fig. SCA analysis for C+A+T+C and C+A+T+E four domains NRPS sequences.** Similar to Fig 3A, we also analyzed 685 C+A+T+C (**A**) and 245 C+A+T+E (**B**) module NRPS sequences by SCA. Although sequence numbers are less than C+A+T composition NRPS (1,161), they also could provide insights for NRPS reengineering. Except known cutting points, the SCA result of C+A+T+E NRPS sequences indicated the junction between A and T domains may be a potential cutting point. Of note, cutting point proposed by Mootz et al.[1]. was for C+A+T+C in original research.
(DOCX)

**S31 Fig. The gap frequency in the MSA of 2,636 A domains.** Related to Fig 6A, upper panel shows the gap frequency in the MSA of 2,636 A domains. Some residues in the substrate-related sectors were found in the highly variable loop regions (L1-L5). These regions contain high numbers of gaps in MSA, and are usually loops in the structure.
(PNG)

**S32 Fig. Clustering and groups of five loops. A.** Hierarchical clustering of the A domains based on the Euclidean distances of their lengths in five loops. A domains were categorized into five groups based on their loop-length vectors. For visual clearness, in calculation, the Euclidean distances which are more than 12 are set as 12 before normalizing. **B.** Loop length profiles of five groups shown in **A**. More details in Method.
(PNG)

**S33 Fig. The entropy and conditional entropy of the specificity-conferring code given different constraints.** The first column shows the entropy of the specificity-conferring code for different substrates. The proteinogenic amino acids are named according to the standard amino acid one-letter code. Abbreviations of non-proteinogenic amino acid substrate: aad = 2-amino-adipic-acid, bht = beta-hydroxy-tyrosine, dab = diaminobutyric acid, dhb = 2,3-dihydroxy-benzoic acid, dhbu = 2,3-dehydroaminobutyric acid, dhpg = 3,5-dihydroxy-phenyl-glycin, horn = hydroxy-L-ornithine, hpg = 4-hydoxy-phenyl-glycine, orn = ornithine and pip = pipecolic acid. Only A domains from 5 main phylum are used to calculate entropy (2564/2623 = 97.8% sequences). The sequence number of each substrate was marked in the bracket after substrate names on the y labels (the first is the number used in the calculation of entropy, and the second is the total number of this substrate in our datasets). Second to fourth columns show the conditional entropy of the specificity-conferring code given information about the phylum, the loop group, and the phylum with the loop group, respectively. Information from the phylum and the loop group could both reduce the uncertainty of the specificity-conferring code, and they together could further reduce the uncertainty.
(PNG)

**S34 Fig. The sequence logo of the specificity-conferring code for substrate alanine in the dimension of phylum and loop group.** 9 of 10 the specificity-conferring code are displayed. The last one is conserved lysine (K) in the A10 motif. It wasn't shown because our A domain sequences only cover A1-A8. Sequence logo will not be plotted, if the number of sequences is less than 3. Substrate abbreviation: A = alanine.
(PNG)

**S35 Fig. The sequence logo of the specificity-conferring code for substrate phenylalanine in the dimension of phylum and loop group.** Similar to S34, but for substrate phenylalanine (F).
(PNG)

**S36 Fig. The sequence logo of the specificity-conferring code for substrate leucine in the dimension of phylum and loop group.** Similar to S34, but for substrate leucine (L).
(PNG)

**S37 Fig. The sequence logo of the specificity-conferring code for substrate valine in the dimension of phylum and loop group.** Similar to S34, but for substrate valine (V).
(PNG)

**S38 Fig. The sequence logo of the specificity-conferring code for substrate tyrosine in the dimension of phylum and loop group.** Similar to S34, but for substrate tyrosine (Y).
(PNG)

**S39 Fig. The sequence logo of the specificity-conferring code for substrate 2-amino-adipic-acid in the dimension of phylum and loop group.** Similar to S34, but for substrate 2-amino-adipic-acid (aad).
(PNG)

**S40 Fig. The sequence logo of the specificity-conferring code for substrate glutamine in the dimension of phylum and loop group.** Similar to S34, but for substrate glutamine (Q).
(PNG)

**S41 Fig. The sequence logo of the specificity-conferring code for substrate diaminobutyric acid in the dimension of phylum and loop group.** Similar to S34, but for substrate diamino-butyric acid (dab).
(PNG)

**S42 Fig. Loop length and group distributions in bacteria and fungi. A.** Comparison loop length between bacteria and fungi in the MiBiG database. The numbers of A domains are 2,370 and 215 for bacteria and fungi, respectively. **B.** Comparison loop length between bacteria and fungi in the larger dataset. The numbers of A domains are 61,494 and 4,484 for bacteria and fungi, respectively. **C.** Loop group distribution in bacteria and fungi in the larger datasets. The numbers of A domains are the same as B. Their loop groups are predicted as the closest one loop group in the MiBiG database by calculating Euclidean distance. A small amount of data (<5%) is not counted because they are the same distance from multiple loop groups.
(PNG)

**S43 Fig. Causal analysis of A domain substrate specificity.** The specificity-conferring code distance used is alignment-score distance. A domain sequence distance used is p-distance. The loop length distance used is Euclidean distance. r is Pearson correlation coefficient. **A.** Causal diagram of A domain substrate specificity. **B.** Relationship between the specificity-conferring code distance and Loop length Euclidean distance. **C.** Relationship between Loop length Euclidean distance and A domain sequence distance. **D.** Relationship between the specificity-conferring code distance and A domain sequence distance. **E.** Relationship between the specificity-conferring code distance and Loop length Euclidean distance for A domains with substrate Ala. Similar to B, but for 430 A domains activating Ala as substrate.
(PNG)

**S44 Fig. Protein sequence pairwise distance distribution. A.** Pairwise distance distribution of 1,161 C+A+T NRPS sequences. The calculation was based on the p-distance, representing the fraction of amino acid being different after global alignment. **B.** Same as that in **A**, but for 685 C+A+T+C NRPS sequences. **C.** Same as that in **A**, but for 245 C+A+T+E NRPS sequences. **D.** Same as that in **A**, but for 2,636 A domain sequences.
(PNG)

**S45 Fig. Workflow of detecting conserved motifs in NRPS domain.** Illustration of the steps in locating known core motifs to query NRPS sequences. First, we curated known core motifs and reference sequences from the literature. Then known motifs on reference sequences were mapped according to previous research, with their locations recorded. Finally, multiple sequence alignment was performed between reference sequences and query sequences. Locations of core motifs in query sequences were inferred by the aligned reference sequences.
(PNG)

**S1 Table. Source of C domain subtype reference sequences.**
(XLSX)

**S2 Table. Subtype prediction by NRPS Motif Finder and annotation by antiSMASH v5 for C domains in MiBiG.**
(XLSX)

**S3 Table. Subtype prediction by NRPS Motif Finder and annotation by antiSMASH v6 for C domains in 16,820 bacterial genomes.**
(XLSX)

**S4 Table. Subtype prediction by NRPS Motif Finder and annotation by antiSMASH v6 for C domains in 2,505 fungal genomes.**
(XLSX)

**S5 Table. Amino acid composition and conservation in three potential motifs.**
(XLSX)

**S6 Table. All available structures of AMP-binding-domain-containing proteins from the PDB database.**
(XLSX)

**S7 Table. Structural effects of point mutations in G-motif G409 of FmqC estimated by Missense3D.**
(XLSX)

**S8 Table. 2,636 A domains sequences information.**
(XLSX)

**S9 Table. Bacterial and fungus genome information used in this study.**
(XLSX)

**S10 Table. The definition of conserved motifs in NRPS domains and the positions in reference sequences.**
(XLSX)

**S1 File. C domain subtype reference HMM files.**
(ZIP)

**S2 File. C domain subtype reference sequences (MSA).**
(ZIP)

**S3 File. The result file of the phylogenetic tree of C domain and E domain by IQ-TREE.**
(ZIP)

**S4 File. NRPS Motif Finder online version code (Python).**
(ZIP)

**S5 File. NRPS Motif Finder Matlab version code.**
(ZIP)

**S1 Text.  Table A, Table B**, **Table C**. Table A. Product yields in different strains by HPLC/MS analysis. Table B. Fungal plasmids and strains used in this study. Table C. PCR primers used in this study.
(DOCX)

## Acknowledgments

We thank Haonan Zheng for helpful discussions. We thank Stefan Günther and Paul F Zierep for providing detailed A domains dataset in SeMPI 2.0. We also appreciated the tech team at the BDA Informatics Suite (www.bdainformatics.org) for the online platform.

## Author Contributions

**Conceptualization:** Zhiyuan Li.

**Data curation:** Ruolin He, Yuanzhe Shao.

**Formal analysis:** Ruolin He, Chen Song.

**Funding acquisition:** Wen-Bing Yin, Zhiyuan Li.

**Investigation:** Wen-Bing Yin, Zhiyuan Li.

**Methodology:** Ruolin He.

**Project administration:** Wen-Bing Yin.

**Software:** Yuanzhe Shao, Long Qian.

**Supervision:** Zhiyuan Li.

**Validation:** Ruolin He, Jinyu Zhang.

**Visualization:** Ruolin He, Jinyu Zhang, Yuanzhe Shao, Shaohua Gu, Long Qian.

**Writing – original draft:** Ruolin He, Jinyu Zhang.

**Writing – review & editing:** Shaohua Gu, Wen-Bing Yin, Zhiyuan Li.

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
