## [Decision Letter · Decision Letter 0]

9 Nov 2022

Dear Dr. Li,

Thank you very much for submitting your manuscript "Non-ribosomal peptide synthetase domain boundary identification and new motifs discovery based on motif-intermotifs standardized architecture" for consideration at PLOS Computational Biology.

As with all papers reviewed by the journal, your manuscript was reviewed by members of the editorial board and by several independent reviewers. In light of the reviews (below this email), we would like to invite the resubmission of a significantly-revised version that takes into account the reviewers' comments.

We cannot make any decision about publication until we have seen the revised manuscript and your response to the reviewers' comments. Your revised manuscript is also likely to be sent to reviewers for further evaluation.

Sincerely,

Turkan Haliloglu

Academic Editor

PLOS Computational Biology

Daniel Beard

Section Editor

PLOS Computational Biology

Reviewer's Responses to Questions

**Comments to the Authors:**

Reviewer #1: Present manuscript from Zhang et al. deals with a very important question in the field of NRPS engineering, namely the identification of domain boundaries and, in addition, discovered new un-described motifs, i.e. within the NRPS adenylation domains (G-motif). The manuscript provides a very comprehensive and detailed overview of the motif and inter-motif architecture of NRPSs and gives a long overdue update on conserved motifs within NRPSs. Additionally, all results are embedded and discussed in the current state-of-the-art and the most important literature.

Although I personally do like this work a lot, because it gives a lot of insights on how to (potentially) improve future engineering campaigns, I also could imagine that this publication is very difficult to comprehend for non-experts in the field. Nevertheless, the NRPS engineering community, including me, will highly appreciate and value this work.

Some minor comments: The first group using A subdomain swaps for NRPS engineering was the Piel group and not Kries et al.. And the Micklefield lab only was able to efficiently introduce (swap) subdomains from highly related origins with very similar and identical specificities. If I recall this correctly the greatest introduced chemical change was Ser/Thr. Even subdomains with the same specificities from different origins yielded low titers.

However, the current manuscript contains many grammatical errors, especially the confusion of singular and plural, missing words and some typos. Even Therefore, the manuscript would benefit from correction by a native speaker.

Reviewer #2: This manuscript represents an attempt to address an important problem in the biosynthesis of non-ribosomally produced peptide natural products. The models that we have to identify and define proteins and domains often do not reflect recent structural and biochemical data, and lack of clarity regarding domain boundaries and regarding the conserved regions involved in important domain-domain interactions make NRPS engineering efforts challenging. Additionally, the complex relationship between the well-established Stachelhaus specificity codes and substrates is frustrating: why do multiple codes result in the binding of one substrate? Why are some codes less specific? In this manuscript, the authors examine NRPS modules (particularly the C and A domains) in depth, attempting to tie specific sequence motifs and loop lengths to altered substrate binding behavior. They also use a set of conserved motifs (as well as sector-based conservation analyses) to better understand domain boundaries. However, based on the data presented, I am not sure the authors have been able to provide enough support for the relevance and broader applicability of the trends that they observe.

Major points of concern:

1. The proposal that a conserved A-domain glycine provides vital flexibility is plausible, and certainly replacement by bulkier amino acids might decrease that flexibility, as suggested in line 343. But the introduction of a bulky, charged residue (G409R) apparently has negligible effect on the production of fumiquinazoline C in the test system; it is not clear how this is reconciled with the authors' model, since the presence of a glycine or even a small uncharged amino acid is apparently not actually "critical" (line 368)? (Given the error bars in Fig. 3F, I'm not sure I'd even confidently say that glycine replacement "always" decreases yield as described on line 654.) Similarly, only two small amino acids were tried. One appeared to have no effect on activity (G409A) and the other (G409P) replaced the most conformationally flexible amino acid (Gly) with the least conformationally flexible amino acid (Pro). While proline replacement did hurt activity, it's not at all clear that this would be true for other smaller amino acids that don't introduce a tertiary backbone amine (Thr, Ser, Cys, Val). The conservation of N397 is also not discussed; if it is not highly conserved, is the proposed steric clash a problem that really can be expected to apply across NRPS domains? Additionally, the authors use a Phyre2 model of FmqC. Even AlphaFold2 urges caution regarding to sidechain conformation, and I haven't seen evidence that Phyre2 outperforms AlphaFold2 in that regard unless there is a close homolog available for modeling (if there is, it is not discussed). What do the equivalent Gly409 and N397 positions in known crystal structures of A domains look like? Several are even referenced in this paper, and quite a few more are available: https://www.acsu.buffalo.edu/~amgulick/NRPSChart.html (In this context, it's also odd that several A-domain subdomains are listed in the SI - fig. S9 and S11 - but never discussed in the context of how the identified motifs in this manuscript might connect to their boundaries.)

2. The domain architecture of FmqC appears to be ATC, while all the analysis in sections 2.1 and 3 focuses explicitly on systems with CAT(-/C/E) architecture. Are differences in domain architecture expected to affect the applicability of any of the results? If not, does limiting the initial analysis to CAT(-/C/E) module sets and not other domain structures artificially affect the analyses by providing a more limited dataset? Similarly, new domains (Cmod, or the interface domain involved in substrate beta-hydroxylation) and additional tailoring domains (methyltransferase, hydroxylase) are not discussed. Since many NRPS systems are more complicated, it would be helpful for the authors to state more clearly whether or not they expect these results to be more broadly applicable, and whether or not similar analyses will need to be made independently for many domain architectures.

3. The cysteine example in Fig. 5C is intriguing, but it seems like it would be quite useful to have equivalent information for all Stachelhaus codes & substrates. It's the kind of information that might help other researchers better understand the relationships between their NRPS A domains of interest and the reasons that they may deviate from established Stachelhaus codes. Unfortunately, I don't think this data is presented anywhere, even in the SI files?

4. While interesting, it is unclear what the differences in average "intermotif" length actually mean in terms of changes to substrate identification in fungal strains. This section isn't really connected to the previous sections, and would really benefit from some discussion about how and why we might expect some of these differences to specifically affect our ability to predict substrates. Even in datasets where the AAD-associated sequences are removed, median and standard deviation seem more informative than the mean: the differences in bacterial and fungal intermotif length in A1-A2 and A7-A8 may be statistically significant in a narrow p<0.05 sense, but it's unclear that they're of any use when considering the architecture of individual proteins, since the vast majority of fungal and bacterial strains appear to have the same intermotif length range, even if there are population-level differences.

5. Two final broad points: first, this work might be stronger if results were validated against larger datasets. MiBIG consists of well-studied biosynthetic pathways, implicitly providing a potential source of bias: understudied and unculturable strains and difficult-to-characterize compounds and the biosynthetic diversity they represent are not present. This may bias analyses. Is similar motif conservation observed across bacterial CAT modules from, say, IMG? Across all bacterial C and A domains in IMG? Whether the answer is yes or no, it will help other researchers understand how widely applicable the conclusions in this manuscript are.

Second, the authors should consider how researchers working with NRPS pathways might be able to apply this data. What would actually be necessary for someone to read this paper and apply the insights to their own NRPS system? The authors don't actually discuss their NRPS Motif Finder tool in the main text or in the SI. Can users search for new sequences or only search for or browse previously evaluated sequences? Which of the analyses discussed will be performed, if it accepts new sequences? Are there expected limitations (will it not handle sequences with alternate domain architectures or additional domains)? A reader for this paper would not know, as it is currently written, and the data in the paper alone don't quite get the reader to the point of being able to apply the results to their system.

Minor issues:

1. Terminology - "Stachelhaus (specificity) code" is the standard terminology. I'm not sure that introducing a new term not really used elsewhere in the vast NRPS literature ("10-S") increases clarity.

2. Details are occasionally lacking in the methods: when sequences with high identity were removed (line 733), how were sets of high identity sequences clustered, and what cutoffs were used to choose a representative one? Where were the 2636 A domain sequences collected from (line 737), and how (they appear to be a mixture of UniProt and MiBIG IDs - are the UniProt sequences via SeMPI?) How and why were the different sector identification parameters defined for CAT, CATE, and CATC NRPS modules (lines 813-820)? Similar issues are present throughout this section. In general, there should be enough detail to reproduce methods, and that detail is sometimes absent.

3. I'd consider using letter or Roman numeral labels rather than color names for the sectors in Fig. 4A and 4C (and in associated discussion, in lines 441-447, and 469). Red/magenta are not very visually distinct, and red and green as adjacent colors will be a problem for people with colorblindness.

4. The colors of sectors 1-6 in Fig. 5A match the phylum colors, but my interpretation of section 4.1 and the caption was not that the sectors directly reflect phylum. I'd strongly suggest an alternate color scheme (or an uncolored but labeled diagram). Additionally, color keys needed to understand Fig. 5B are on the opposite end of the figure, and they look like they are associated with Figs. 5A/C. I'd try to redesign this figure for clarity.

5. "Light green" or "yellow green" would likely be a more widely recognized color term than "limon" (line 480, SI)

6. Finally, there are numerous small errors (aspartic acid referred to as positively charged (Fig 2), lysine and arginine described as negatively charged (Fig 2, Table 1), typos like UniPort (line 737). An additional readthrough to catch and correct this sort of error would be advisable.

Reviewer #3: Please see attached document

**Have the authors made all data and (if applicable) computational code underlying the findings in their manuscript fully available?**

Reviewer #1: Yes

Reviewer #2: **No: **Not in this draft. For certain sorts of data (MSAs, matrices, data used to generate heat maps), nothing seems to be available at all beyond the figures. And it's implied that some sorts of data exist (Fig. 5C shows data connecting substrate, Stachelhaus code, and loop length) but it is not presented elsewhere or for other substrates. The source code was described as being in the SI (line 892) but didn't seem to be present in the SI files I downloaded? Additionally, the data points behind means and medians - in Fig. 6 and its SI equivalent, or Fig. 3F - definitely didn't seem to be present in any of the 3 Excel files.

Reviewer #3: Yes

PLOS authors have the option to publish the peer review history of their article (what does this mean?). If published, this will include your full peer review and any attached files.

Reviewer #1: No

Reviewer #2: No

Reviewer #3: No
---

## [Decision Letter · Decision Letter 1]

14 Mar 2023

Dear Dr. Li,

Thank you very much for submitting your manuscript "Knowledge-guided data mining on the standardized architecture of NRPS: subtypes, novel motifs, and sequence entanglements" for consideration at PLOS Computational Biology. As with all papers reviewed by the journal, your manuscript was reviewed by members of the editorial board and by several independent reviewers. The reviewers appreciated the attention to an important topic. Based on the reviews, we are likely to accept this manuscript for publication, providing that you modify the manuscript according to the review recommendations.

Sincerely,

Turkan Haliloglu

Academic Editor

PLOS Computational Biology

Daniel Beard

Section Editor

PLOS Computational Biology

Reviewer's Responses to Questions

**Comments to the Authors:**

Reviewer #1: Zhang et al. provide a very comprehensive overview of conserved NRPS domain motifs and inter motif sequences. However, compared to the initial manuscript the revised version did not substantially improve. Nevertheless, I value the authors work, because I am still convinced that it provides a well overdue update on NRPSs.

Prior to publication the following points must be solved:

Major comments:

- Fig. 3 & 4 are the same.

- Caption of Fig. 3 doesn't match or vice versa

Minor comments:

- The manuscript would still benefit from correction by a native speaker - there are still too many mistakes. Some parts obviously improved but others did not.

Reviewer #2: The restructuring and addition of additional background have substantially improved the clarity and accessibility of the paper. It's also very reassuring to see that some of the extended analyses on larger datasets (or analyses using alternate methods, e.g. Phyre2 vs. SWISS-MODEL vs. AlphaFold2) have validated (and even strengthened) some of the authors' conclusions! In its revised form, I think the manuscript is significantly more compelling, and it will be much easier for other groups to apply its conclusions to their attempts to investigate and/or engineer NRPS systems. However, a few issues still remain in this draft.

Major issues:

1. Regarding the G-motif: the preliminary data regarding the effect of module position in multimodule pathways are quite intriguing! However, if those data are not included in this manuscript, it unfortunately remains premature to describe the G-motif as "critical" or that the flexibility of glycine at that position is "essential" (line 853) - despite the high conservation of that residue, what's going on is clearly more complex than an absolute reliance on a glycine at that position for activity. Even the preliminary data in in Additional Figure 3 don't show that a glycine at any position is actually vital for enzyme activity (even in Module 1, other small amino acids are tolerated with a moderate hit to activity). Additionally, statements in the paper should describe the data presented in the paper, and in this context, given the data that are in the paper (specifically the G409A and G409R results in Fig. 4), I do not think it is appropriate to say "Our mutation experiment supported the critical role of the conserved glycine in the G-motif, and indicated that the flexibility of glycine is important in this position" (less flexible residues, including those with different chemical properties, retained high activity.) Similarly, it's premature to say that "Our results implied that the flexibility of glycine in the G-motif is essential for conformational change during A domain functions, therefore conformational restrictions in the G-motif impede A domain function." It's quite plausible that that is part of what's happening (and part of the reason for the different results in multimodule systems), but the data actually in this paper (diminished natural product production for a set of variants in a single module in a single NRPS system) do not provide direct evidence for the broader mechanism of G-motif function. More conservative and less definitive statements regarding the G-motif remain appropriate.

2. Some details remain missing in the methods section. E.g. a maximum likelihood tree for the condensation domain is shown in Fig. 2, but the alignment and parameters (and even the tool) used to generate it are not actually described. Structural modeling and alignment are mentioned in the text and in the SI but not in the methods (and it's unclear how the "simulated mutation" was performed, and what methods were used to explore tryptophan sidechain conformations in this simulation). It's fine to cite previous papers where appropriate, but all methods do need to be described in the methods section, either in experimental detail or as citations.

3. Figure 3 appears to be the wrong figure entirely (it seems to be a duplicate of figure 4, and the captions don't match the content at all.)

Minor issues:

1. Many paragraphs are quite long. Splitting them at natural points (e.g. in line 76, the paragraph switches from describing NRPS modules to discussing engineering efforts) would improve readability.

2. The captions in Fig. 1 should clarify colors for all violin plots. It looks like C must be grey, A must be blue, T is yellow, maybe E is grey (hard to see at printed scale)?, TE is red? But this is not stated except for the first violin plot.

3. In Fig. 2: I'd recommend using red and blue rather than red and green for bacteria vs. fungi for colorblind readers. The black tickmarks under each position in the logos overlap the letters in ways that might impede legibility. It's unclear what "Three NRPS organizations do not have known examples" means - is it that these NRPS layouts are observed in BGCs in genomes but none of the products that they produce have yet been structurally characterized? (In context, it's a little unclear, since it could also be interpreted as "these layouts could theoretically exist but have not been observed in genomes")

4. In section 1.5, it might be worth noting what percentage of NRPS pathways actually contain only high-confidence C domains, i.e. how big was the dataset from which these were extracted and what percentage of the dataset is being analyzed in the end. (The conclusions risk being a little circular: the high-confidence NRPS clusters are dominated by well-studied types of siderophores because those systems are, well, well-studied.) It would help to clarify for the reader how representative these systems actually are so that they don't overapply conclusions to the larger NRPS dataset.

5. Line 577-578: "We observed that two amino acids, N577 and F493 in LgrA, have a size near glycine in the G-motif (G505)." It's unclear what this is saying? Asn and Phe aren't similar in size to Gly and aren't in the G-motif specified in the prior page. Rephrase for clarity?

6. Line 688: It's confusing to find Glu grouped together with "other conserved aliphatic and aromatic amino acids" (it is neither) that form the hydrophobic core of the domain? I'd clarify the description or its potential interactions, since it's a key part of this motif.

7. Line 1181 - more detail is appropriate here (e.g. what was the LC gradient? What was the mobile phase? Were data collected in positive or negative ion mode, and what were the other mass spec parameters?) Again, the methods do need to be available somewhere, as a citation or as part of the methods section. (If the LC method is the method described in lines 1207-1214, I'd put all that information in one place.)

8. References - format varies. (Sentence case vs. All Capitalized Words, sometimes species names and gene names are not italicized, journal names are sometimes abbreviated and sometimes not (and sometimes capitalized and sometimes not), etc.) These need to conform to the PLOS reference style for publication (see: https://journals.plos.org/ploscompbiol/s/submission-guidelines)

Typos:

There remain numerous small errors in grammar and spelling. The manuscript would benefit from a careful editing pass. I've listed some examples but this should not be assumed to be a complete list:

1. "boarders" rather than "borders" in Fig. 1

2. Inconsistent description of sequence conservation - PXXGXXYF vs. PxxGxGYG (capital vs. lower case X)

3. obtained from "literatures" rather than "the literature" in line 354.

4. "ester- and amide- bond" rather than "ester and amide bonds" in line 363

5. "assigns" not "assign" in line 390 - antiSMASH is probably best thought of as singular in this context.

6. "lichenysins synthase" and "depsipeptides or coprogens synthetase" - in both cases the plurals should be on "synthetase"

7. "top 29th in organizations" in line 472-3 - should probably be "29th most common organization"? ("top (number)" is usually used for a range of numbers, not a single one)

8. ε, not e in "poly-e-Lys" in line 591

9. "Both" is inappropriately capitalized in line 743 (and "work of XUC" and "work of XU" in that line and the next should be rephrased - it's contextually unclear what XUC and XU are, and I might try to more explicitly state what the product of the pathway was and the module layout, rather than just mentionining the sector affected by the cut site.)

10. "2+" should be a superscript in line 777

11. "dhb" in Fig. 6 vs. "Dhb" in line 853 vs. lowercase again in Fig. S33

12. "NRPS undergoes" - rephrase, probably "NRPS complexes undergo substantial..."

13. line 1062 - should be "from" not "form"

14. line 1072 - should be "Clustal" not "Cluster"

15. line 1173 - "Escherichia coli" should be italicized.

16. line 1080 - is "3 mm" supposed to be a micron? (That is, is it referring to 3µm particle size?)

Note that these issues extend to SI material, e.g. "Hybird PKS-NRPS" (not "hybrid") in Table S1 or "they in fungi" in line 215 & 216 of the SI text (I think these lines are saying that the numbers of bacterial and fungal sequences differ for the A1-Tα1 and Tα1-T1 analyses for various reasons? Note that this should also be fixed in the Fig. S14 caption). Similarly in line 243, it should probably "the bL subtype" not "bL the subtype." And in Fig. S42, the plot titles should contain the word "between" not "betweem", Fig. S45 should refer to "the literature" or "publications" or "published data" or something along those lines, not "literatures", and the SI references should be correctly formatted as well.

Reviewer #3: The authors addressed all the concerns in this revised version.

**Have the authors made all data and (if applicable) computational code underlying the findings in their manuscript fully available?**

Reviewer #1: Yes

Reviewer #2: Yes

Reviewer #3: Yes

PLOS authors have the option to publish the peer review history of their article (what does this mean?). If published, this will include your full peer review and any attached files.

Reviewer #1: No

Reviewer #2: No

Reviewer #3: No

Figure Files:

Data Requirements:

Reproducibility:

References:

---

## [Editor Report · Decision Letter 2]

12 Apr 2023

Dear Dr. Li,

We are pleased to inform you that your manuscript 'Knowledge-guided data mining on the standardized architecture of NRPS: subtypes, novel motifs, and sequence entanglements' has been provisionally accepted for publication in PLOS Computational Biology.

Best regards,

Turkan Haliloglu

Academic Editor

PLOS Computational Biology

Daniel Beard

Section Editor

PLOS Computational Biology

---

## [Editor Report · Acceptance letter]

9 May 2023

PCOMPBIOL-D-22-01247R2 

Knowledge-guided data mining on the standardized architecture of NRPS: subtypes, novel motifs, and sequence entanglements

Dear Dr Li,

I am pleased to inform you that your manuscript has been formally accepted for publication in PLOS Computational Biology. Your manuscript is now with our production department and you will be notified of the publication date in due course.

With kind regards,

Timea Kemeri-Szekernyes
